# NeuRodin: A Two-stage Framework for High-Fidelity Neural Surface Reconstruction

Yifan Wang[1,2]    Di Huang[2]    Weicai Ye[2,3,✉]    Guofeng Zhang[3]    Wanli Ouyang[2]    Tong He[2,✉]

[1]Shanghai Jiao Tong University    [2]Shanghai Artificial Intelligence Laboratory
[3]State Key Lab of CAD&CG, Zhejiang University

maikeyeweicai@gmail.com    tonghe90@gmail.com

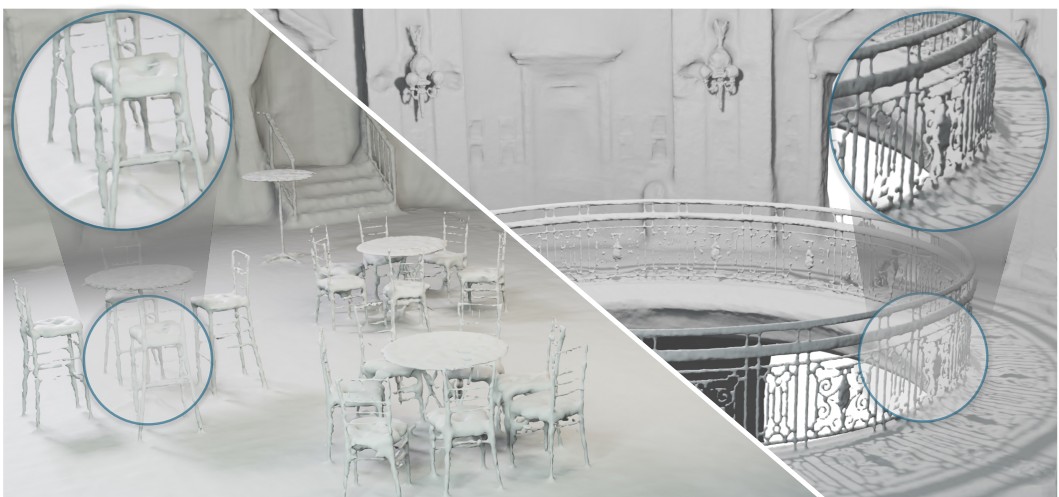

Figure 1: We present **NeuRodin**, a novel two-stage framework designed for high-fidelity neural surface reconstruction with intricate structures. Requiring only posed RGB captures as inputs, **NeuRodin** not only recovers large-scale areas but also accurately reconstructs fine-grained details.

## Abstract

Signed Distance Function (SDF)-based volume rendering has demonstrated significant capabilities in surface reconstruction. Although promising, SDF-based methods often fail to capture detailed geometric structures, resulting in visible defects. By comparing SDF-based volume rendering to density-based volume rendering, we identify two main factors within the SDF-based approach that degrade surface quality: *SDF-to-density representation* and *geometric regularization*. These factors introduce challenges that hinder the optimization of the SDF field. To address these issues, we introduce **NeuRodin**, a novel two-stage neural surface reconstruction framework that not only achieves high-fidelity surface reconstruction but also retains the flexible optimization characteristics of density-based methods. **NeuRodin** incorporates innovative strategies that facilitate transformation of arbitrary topologies and reduce artifacts associated with density bias. Extensive evaluations on the Tanks and Temples and ScanNet++ datasets demonstrate the superiority of **NeuRodin**, showing strong reconstruction capabilities for both indoor and outdoor environments using solely posed RGB captures. Project website: https://open3dvlab.github.io/NeuRodin/

38th Conference on Neural Information Processing Systems (NeurIPS 2024).

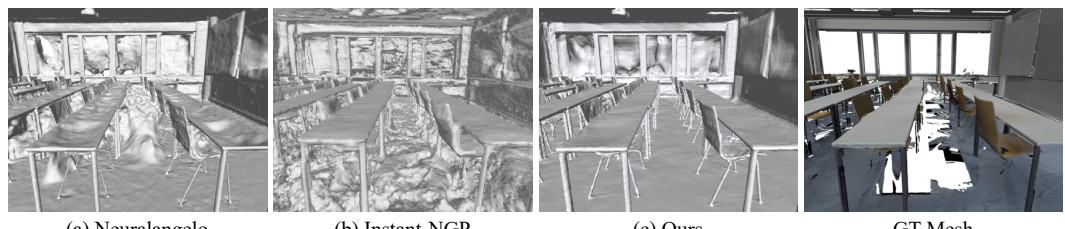

| (a) Neuralangelo | (b) Instant-NGP | (c) Ours | GT Mesh |

Figure 2: **Comparative analysis of SDF-based and density-based volume rendering methods.**
**(a)** Neuralangelo [13] experiences difficulties with topological transformations, leading to incorrect surfaces. **(b)** Instant-NGP [18] approximates the correct surface positioning yet produces a noisy surface. **(c)** Our method achieves high-quality surfaces with fine details.

## 1 Introduction

3D surface reconstruction [34, 16, 35, 28, 8, 24, 26, 40, 39, 14, 12, 2, 38, 25] is a long-standing research topic in the field of computer vision. This process involves using images to recover the underlying 3D geometry, typically represented as meshes. These reconstructed meshes find diverse applications in various domains, including video games and augmented/virtual reality systems. In this paper, we specifically address the challenge of reconstructing 3D surfaces from posed RGB images.

Inspired by the density-based representation [15] for task of novel view synthesis, recent works for neural surface reconstruction commonly introduce signed distance functions (SDF) [28, 35] to recover high-quality geometry.

However, incorporating SDF to the density function is nontrivial and often fails to intricate geometric details. We illustrate this by comparing two methods for reconstructing the same scene: *Instant-NGP* [18], which employs density-based volume rendering, and *Neuralangelo* [13], which utilizes SDF-based volume rendering. Both methods use a similar multi-resolution hash table representation. The mesh is produced by is by TSDF-fusion [4] for *Instant-NGP*. As illustrated in Figure 2, *Instant-NGP* reconstructs surfaces with accurate localization albeit with a certain roughness, while *Neuralangelo* produces smoother surfaces yet encounters issues in correctly positioning portions of the surface. This disparity underscores the limitations and highlights the need for improved modeling capabilities in SDF-based surface reconstruction methods.

*Why do SDF-based surface reconstruction methods face challenges in accurately capturing intricate geometric details, and how can these methods be improved?* In this paper, we thoroughly analyze the reconstruction pipeline and identify two primary factors in current SDF-based pipelines that contribute to suboptimal surface reconstruction:

- *SDF-to-density conversion:* SDF-based volume rendering requires a conversion function to relate the SDF field with the density field. Existing methods use a conversion function that assigns uniform density across the same level sets, which restricts the representation of arbitrary non-negative density values. Additionally, there is no assurance that the geometric representation within a volume rendering framework will align perfectly with the implicit surface. This misalignment often results in accurate visual renderings on incorrect surfaces, due to inherent biases.

- *Geometric regularization:* Regularization constraints imposed on the implicit surface can limit topological changes during optimization. These constraints often introduce biases, complicating the convergence of the model and hindering its ability to accurately represent complex geometries.

To tackle these challenges, we introduce NeuRodin (Figure 1), a high-fidelity 3D surface reconstruction method that innovatively overcomes the limitations previously outlined. Firstly, we refine the SDF-to-density conversion by transitioning from a global scale parameter to a local adaptive parameter. Unlike previous methods that enforce the same densities for points with identical SDF values, our approach enhances the flexibility and effectiveness of the SDF function by allowing adaptive density values. Secondly, we implement a novel loss function designed to align the maximum probability distance with the zero-level set in volume rendering, improving the alignment of geometric representations. Additionally, We incorporates above innovations within a two-stage optimization framework to tackle the over-regularization imposed by geometric constraints. Initially, we employ a coarse optimization stage in which the SDF field operates similarly to a density field,

exhibiting minimal influence from topological transformations. Subsequently, a refinement stage is conducted to achieve a surface with enhanced smoothness. We also introduce the stochastic-step numerical gradient estimation technique to mantain a natural zero level set for the coarse stage. With the design described, our method enables high-fidelity surface reconstruction suited to both large-scale and intricate geometries.

We conducted extensive experiments on the Tanks and Temples dataset [11] and the ScanNet++ dataset [41], where our method demonstrated superior performance over the previous state-of-the-art in both indoor and outdoor environments. Notably, recognizing the lack of an established benchmark for ScanNet++, we executed comparative analyses using six baseline methods and established a new benchmark for ScanNet++ reconstruction, significantly enriching the community resources and setting a foundation for future research. In comparative tests, our model outperformed Neuralangelo on the Tanks and Temples training set, delivering superior results with only 1/8 the parameters of the comparative model. Our approach excels in optimizing complex topological structures and preserving intricate details, enabling high-fidelity, fine-grained surface reconstruction.

## 2    Related Work

**Multi-view 3D reconstruction.** In traditional 3D surface reconstruction, methods based on Multi-View Stereo (MVS) have long been prevalent, serving as a foundational approach for mining sparse geometric data from multiple views and generate detailed 3D models by comparing and analyzing the disparities across multiple camera perspectives. The traditional MVS methods [24], while effective in texture-rich domains, often stumble upon the challenge of processing ambiguous observations. The point clouds produced by traditional MVS methods suffer from noise, undermining the reliability of the surface triangle meshes reconstructed from these point clouds. For learning-based MVS methods [34, 44, 33, 32], the generated point clouds are still plagued by noise, leading to consistently incomplete reconstructions.

**Neural surface reconstruction.** NeRF [16, 37, 10, 17] pioneers the use of neural network to represent neural radiance fields for novel view synthesis and optimizes these scenes through differentiable volume rendering. Following NeRF, subsequent research has combined implicit surfaces with differentiable volume rendering [19, 28, 35]. These methods typically represent implicit surfaces as SDF and use the zero-level set of SDFs to describe geometry, achieving high-quality reconstruction on individual objects. Various improvements have been made based on this foundation, including the incorporation of different positional encoding to enhance representational capabilities [22, 29, 47, 48] and the introduction of additional priors to deal with surfaces that exhibit specular highlights or have low textures [43, 27]. Several studies refine the modeling of SDF-to-density conversion [46, 31] to address bias issues in density. Meanwhile, other works employ patch-match techniques to improve multi-view consistency [5, 7]. Neuralangelo [13] enhances the network's representational capability by introducing hash encoding. Additionally, it proposes numerical gradients and coarse-to-fine optimization strategies to enhance the quality of surface reconstruction.

## 3    Study on SDF-based Volume Rendering

### 3.1    Preliminary

**Density-based volume rendering**    Density-based volume rendering methods model a 3D scene as a volume density field. Given a camera position $\mathbf{o}$ and view direction $\mathbf{d}$, the ray emitted from $\mathbf{o}$ in direction $\mathbf{d}$ is denoted as $\{\mathbf{r}(t) = \mathbf{o} + t\mathbf{d} | t > 0\}$. A set of $n$ points is sampled along this ray. The predicted density $\sigma(\mathbf{r}(t))$ and geometry features $\mathbf{z}(\mathbf{r}(t))$ of the point $\mathbf{r}(t)$ are obtained from a geometry network $\phi_{\text{geo}}$. the density is parameterized by an activation function, such as `ReLU`, `softplus`, or the `exp` function, prior to being output by the network.

A color network $\phi_{\text{color}}$ predicts the color $\mathbf{c}(\mathbf{r}(t), \mathbf{d})$, taking as inputs the geometry feature $\mathbf{z}(\mathbf{r}(t))$, and the viewing direction $\mathbf{d}$. The rendered color of this ray can be calculated as:

$$\hat{C}(\mathbf{r}) = \int_0^{+\infty} T(t)\sigma(\mathbf{r}(t))\mathbf{c}(\mathbf{r}(t)), \quad T(t) = \exp\left(-\int_0^t \sigma(\mathbf{r}(u))du\right). \tag{1}$$

The networks are trained to minimize a color loss $\mathcal{L}_{\text{color}}$ that quantifies the difference between the rendered colors and the ground truth colors:

$$\mathcal{L}_{\text{color}} = \frac{1}{m} \sum_{\mathbf{r} \in \mathcal{R}} \|\hat{C}(\mathbf{r}) - C(\mathbf{r})\|_1, \tag{2}$$

with $\mathcal{R}$ representing the set of $m$ rays in each training batch and $C(\mathbf{r})$ being the ground-truth color of ray $\mathbf{r}$.

**SDF-based volume rendering**  SDF-based volume rendering methods, such as NeuS [28] and VolSDF [35], combine volume rendering with an SDF representation. Unlike density-based methods, the geometry is represented as the zero level set of the SDF. The predicted SDF $f(\mathbf{r}(t))$ and geometry feature $\mathbf{z}(\mathbf{r}(t))$ at the point $\mathbf{r}(t)$ are obtained from the geometry network $\phi_{\text{geo}}$. Then, the SDF $f(\mathbf{r}(t))$ is transformed into density $\sigma(\mathbf{r}(t))$ by a predefined function $\Psi$ and a global scale $s$. For instance, VolSDF defines the density as the scaled cumulative distribution function of the negative SDF:

$$\sigma(\mathbf{r}(t)) = \Psi_s(f(\mathbf{r}(t))) = \begin{cases} \frac{1}{2s} \exp\left(\frac{-f(\mathbf{r}(t))}{s}\right) & \text{if } f(\mathbf{r}(t)) \geq 0, \\ \frac{1}{s}\left(1 - \frac{1}{2}\exp\left(\frac{f(\mathbf{r}(t))}{s}\right)\right) & \text{if } f(\mathbf{r}(t)) < 0. \end{cases} \tag{3}$$

Moreover, to encourage the SDF to have a unit norm gradient, the Eikonal loss [9] is often employed:

$$\mathcal{L}_{\text{eik}} = \frac{1}{mn} \sum_{\mathbf{r},t} \left(\|\nabla f(\mathbf{r}(t))\| - 1\right)^2. \tag{4}$$

Incorporating this loss not only helps to avoid suboptimal solutions at the zero level set but also promotes smoothness. The networks are trained under the supervision of both the color loss $\mathcal{L}_{\text{color}}$ and the Eikonal loss $\mathcal{L}_{\text{eik}}$.

## 3.2  Challenges in Previous SDF-Based Volume Rendering

State-of-the-art SDF-based volume rendering techniques frequently fail to reconstruct surfaces with accuracy in scenarios where density-based methods manage to renders realistic novel views. This disparity highlights the inherent limitations of SDF-based volume rendering approaches. To further elucidate these issues, we explore the fundamental distinctions between SDF-based and density-based volume rendering. Please refer to the appendix for a more in-depth analysis.

**Unsuitable assumption for SDF-to-density conversion.** SDF-based volume rendering methods typically employ a predefined function $\Psi$ and a global scale parameter $s$ to convert SDF values into density, as described by Equation (3). These methods often result in uniform density values for points sharing identical SDF values. Such a global scaling mechanism restricts the representation capability of the density field derived from the SDF field. Intuitively, previous top-performing methods for novel view synthesis can generate arbitrary non-negative density values within $(0, +\infty)$. In contrast, incorporating SDF representation with a global scaling factor for surface reconstruction can only result in density values within $\left(0, \frac{1}{s}\right]$.

**Bias of the density.** When applying an Eikonal constraint or any form of smooth regularization to an SDF, the geometric representation within the rendering framework must align with that of the SDF. Unfortunately, current SDF-based methods often fail to ensure this alignment, particularly at larger-scale parameters $s$ as explored mathematically in [46] to analyze this issue. Recent studies [31, 46, 3] have attempted to tackle this issue, proposing designs for SDF to density conversion that aim to minimize bias. Despite these advancements, these solutions still exhibit inherent biases. Additionally, the introduction of geometric regularization often exacerbates this bias, complicating model convergence and resulting in the creation of inaccurate surfaces. A more detailed analysis of this issue is provided in Section 4.2 and Appendix B to Appendix D.

**Over-regularization of Geometry.** To maintain a high-quality surface, previous methods often introduce geometric constraints, such as Eikonal loss or smoothness constraints. However, these global constraints result in excessive smoothing across all regions, both flat and intricate, leading to a loss of fine details. Moreover, in the framework of SDF-based volume rendering, the prediction of color typically necessitates being conditioned on normals following IDR [36], a characteristic that distinctly sets it apart from the density-based volume rendering approach. When optimizing the

color conditioned on the normal and explicitly constraining the SDF by geometric regularization, the optimization process restricts the topological structure. Please refer to the Appendix F for the impact on the normal condition.

## 4 Method

### 4.1 Uniform SDF, Diverse Densities

To deal with the representation limitations of SDF-transformed density, instead of using a global scale $s$ for the transformation from SDF to density, we have employed a strategy akin to that of [30], which utilizes a non-linear mapping to obtain the unique scale $s$ associated with a given point $\mathbf{r}(t)$. More precisely, it is defined as follows:

$$(f(\mathbf{r}(t)), s(\mathbf{r}(t)), \mathbf{z}(\mathbf{r}(t))) = \phi_{\text{geo}}(\mathbf{r}(t)), \quad \sigma(\mathbf{r}(t)) = \Psi_{s(\mathbf{r}(t))}(f(\mathbf{r}(t))). \tag{5}$$

With this particular design, the density is not identical within the same SDF level set and can achieve any non-negative value through the continuous representation that maps an input coordinate to its corresponding scale.

This approach ensures that densities within the same SDF level set are no longer uniformly identical. Instead, they can vary, achieving any non-negative value through a continuous representation that maps input coordinates to their corresponding scales. This design greatly enhances the flexibility and accuracy of our density modeling, enabling more realistic and detailed reconstructions. More detailed analysis regarding the local scale can be found in Appendix A.

### 4.2 Explicit Bias Correction

The issue of bias represents a critical concern frequently addressed within SDF-based volume rendering. As demonstrated in Figure 3, it is necessary to align the geometric representation under the volume rendering framework with that of the implicit surface. For the volume rendering framework, the most intuitive way to represent geometry is through the *rendered distance*:

$$\hat{D}_{\text{rendered}}(\mathbf{r}) = \int_0^{+\infty} T(t)\sigma(\mathbf{r}(t))t \, dt. \tag{6}$$

We can also consider the position where $w(t)$ is maximized — that is, the probability that the light ray arrives and collides is the greatest, or in other words, the location that contributes the most to the color — as the geometric representation within the volume rendering framework:

$$\hat{D}_{\text{prob}}(\mathbf{r}) = \arg\max_{t \in (0, +\infty)} w(t) = \arg\max_{t \in (0, +\infty)} T(t)\sigma(\mathbf{r}(t)). \tag{7}$$

We shall refer to $\hat{D}_{\text{prob}}(\mathbf{r})$ as the *maximum probability distance*. For an implicit surface, *the zero level set* offers a direct geometric representation. In an ideal scenario, irrespective of whether convergence has been achieved, the geometric representations of volume rendering (i.e., *rendered distance* and *maximum probability distance*) and the geometric representation of the implicit surface (i.e., *zero level set*) should be aligned, as illustrated in Figure 3 (a). However, in the practical optimization process, conflicts such as those depicted in Figure 3 (b) may arise, leading to misalignment between the two representations.

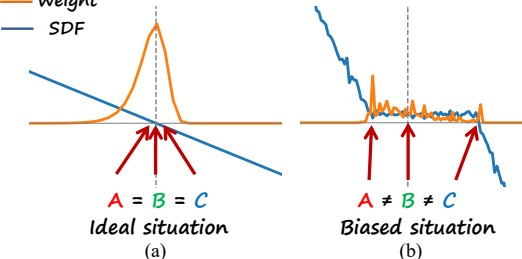

Figure 3: **Visualization of the bias of the density.** (a) An ideal scenario where the geometry of the volume rendering scheme (*A: maximum probability distance* and *B: rendered distance*) aligns precisely with the geometry of the implicit surface (*C: zero level set*). (b) A biased scenario showcasing misalignment.

Multiple past methods have broached this topic, offering various solutions. For example, in TUVR [46], an unbiased model is proposed and it is mathematically proven that the zero-crossing point of the SDF is at a local maximum of rendering weight. However, bias still exists. The experimental analysis is presented in Section 5.3. In the work

of [3], a penalty is imposed on the rendered distance to ensure that its SDF value is equal to zero. However, it is important to note that the rendered distance is subject to significant distortion due to existing biases, particularly in the early stages of convergence as shown in Figure 3 (b). We offer further analysis from Appendix B and Appendix D.

We propose an explicit bias correction in which we opt to deliberately align the maximum probability distance with the zero level set. Specifically, we define

$$\mathcal{L}_{\text{bias}} = \frac{1}{m} \sum_{\mathbf{r} \in \mathcal{R}} \max\left(f(\mathbf{r}(t^* + \epsilon_{\text{bias}})), 0\right), \quad t^* = \arg\max_{t \in (0, +\infty)} T(t)\sigma(\mathbf{r}(t)), \tag{8}$$

where $\epsilon_{\text{bias}}$ is a bias correction factor. The loss function is designed to penalize the positive portion of $f(\mathbf{r}(t^* + \epsilon_{\text{bias}}))$, which encourages the SDF to take on negative values after the maximum probability distance. We have shown in Appendix C that manually scheduling the lower bound of the local scale, coupled with the penalty after the point of maximum probability distance, can effectively alleviate bias issues. During the experiment, we approximate $t^*$ by directly using the sampled point with the largest $w(t)$, which, despite introducing a certain degree of deviation, does not affect the overall effectiveness. Please refer to Appendix C for details on the design.

### 4.3 Two-Stage Optimization to Tackle Geometry Over-Regularization

Previous methods often produce incorrect surfaces due to over-regularization of geometry as shown in Figure 2 (a). However, we have discovered that methods based on density are not constrained by changes in topology as shown in Figure 2 (b), prompting us to question whether the SDF field can be first optimized as freely as density field, then refine to a smooth surface by geometry regularization. We now propose a novel two-stage optimization approach. This approach allows the optimization process to initially mimic density-based behavior in the first stage and subsequently refines to a smooth surface in the second stage.

For the first stage, our objective is to tackle the over-regularization issue. An intuitive solution might be to eliminate or downweight any geometric constraints and avoid conditioning the color on the predicted normal, but this approach often results in an unnatural zero level set [9, 28, 35, 36]. We experimentally validated in the Appendix H.6.

We have identified a simple but effective method to preserve the natural level sets of large-scale structures while allowing the formation of complex structures to be unimpeded by geometric regularization. Instead of applying geometric regularization directly to the gradient $\nabla f(\mathbf{r}(t))$, we elect to impose them upon an estimated gradient $\hat{\nabla} f(\mathbf{r}(t))$, to which we introduce uncertainty through a specific design. Specifically, the $x$-component of the estimated gradient is

$$\hat{\nabla}_x f(\mathbf{r}(t)) = \frac{f(\mathbf{r}(t) + \boldsymbol{\epsilon}_x) - f(\mathbf{r}(t) - \boldsymbol{\epsilon}_x)}{2\epsilon}, \quad \text{where } \boldsymbol{\epsilon}_x = (\epsilon, 0, 0) \text{ and } \epsilon \sim U(0, \epsilon_{\text{max}}). \tag{9}$$

The gradient is calculated through finite differences similar to those described in [13, 21]. However, the step size for gradient estimation at each iteration is stochastically sampled from a uniform distribution in the range of $(0, \epsilon_{\text{max}})$.

Using this technique, we observed that estimated larger-scale normals have smaller variance, while fine details exhibit larger variance, as depicted in Figure 4. This introduces uncertainty in geometric regularization, ensuring stability for large features and flexibility for complex details. We provide further explanation in Appendix E.

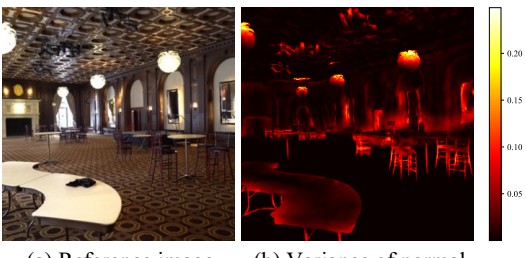

(a) Reference image     (b) Variance of normal

Figure 4: The heatmap for the variance of the normal predicted using random step.

During the initial stage, our goal is to reconstruct the approximate, coarse structure of the 3D content. This is primarily addressed by tackling the issues of over-regularization and the bias in density estimation that we previously mentioned. We employ the stochastic-step numerical gradient estimation along with the explicit bias correction to address the initial reconstruction of the coarse

| Metric | F-Score ↑ | | | | | | | | |
|---|---|---|---|---|---|---|---|---|---|
| Scene | NeuralWarp | COLMAP | NeuS | Geo-NeuS | NeuS-NGP | MonoSDF | Neuralangelo* | Neuralangelo | Ours |
| Barn | 0.22 | 0.55 | 0.29 | 0.33 | 0.46 | 0.49 | 0.61 | **0.70** | **0.70** |
| Caterpillar | 0.18 | 0.01 | 0.29 | 0.26 | 0.32 | 0.31 | 0.34 | **0.36** | **0.36** |
| Courthouse | 0.08 | 0.11 | 0.17 | 0.12 | 0.08 | 0.12 | 0.13 | **0.28** | 0.21 |
| Ignatius | 0.02 | 0.22 | 0.83 | 0.72 | 0.81 | 0.78 | 0.82 | **0.89** | 0.87 |
| Meetingroom | 0.08 | 0.19 | 0.24 | 0.20 | 0.08 | 0.23 | 0.22 | 0.32 | **0.43** |
| Truck | 0.35 | 0.19 | 0.45 | 0.45 | 0.44 | 0.42 | 0.45 | **0.48** | 0.47 |
| Mean | 0.15 | 0.21 | 0.38 | 0.35 | 0.37 | 0.39 | 0.43 | 0.50 | **0.51** |

Table 1: **Quantitative evaluation of our method on the Tanks and Temples training subset.** The **best** performance and the second-best outcomes are highlighted for easy reference. Note that the hash grid parameters used in our method is the same as Neuralangelo*, which possesses $2^{19}$ hash entries per resolution.

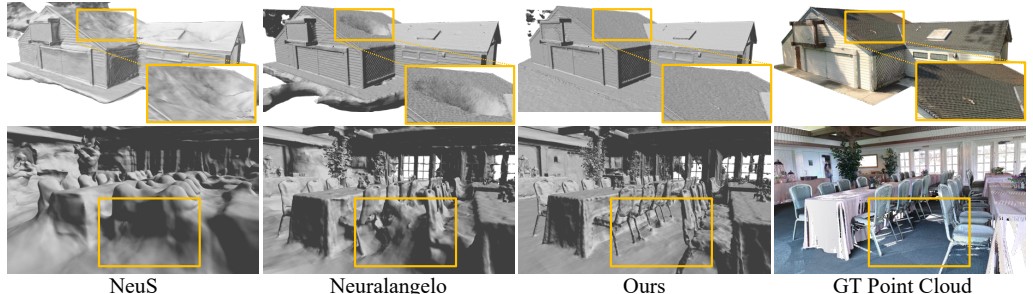

NeuS          Neuralangelo          Ours          GT Point Cloud

Figure 5: **Quantitative comparison on the training subset of Tanks and Temples dataset.**

structure of the 3D content. Additionally, we utilize VolSDF's SDF-to-density conversion, with the local scale modeling delineated in Equation 5, to facilitate this primary formulation. Consequently, the training loss at this stage is formulated as:

$$\mathcal{L}_{\text{coarse}} = \mathcal{L}_{\text{color}} + \lambda_{\text{eik}}\mathcal{L}_{\text{eik}}(\hat{\nabla} f) + \lambda_{\text{bias}}\mathcal{L}_{\text{bias}}. \tag{10}$$

During the refinement stage, we discontinue the use of the estimated gradients given that the fundamental 3D content has been initially restored and the issue of over-regularization no longer presents a concern. In a similar vein, we move past the explicit bias correction, as the significant surface errors induced by bias were initially addressed in the first stage. We incorporate a standard Eikonal loss alongside a smoothness constraint from PermutoSDF [22] to enforce local smoothness:

$$\mathcal{L}_{\text{smooth}} = \frac{1}{mn}\sum_{\mathbf{r},t}\left(\mathbf{n}\left(\mathbf{r}(t)\right)\cdot\mathbf{n}\left(\mathbf{r}(t) + \epsilon_s\boldsymbol{\eta}(\mathbf{r}(t))\right) - 1\right)^2, \tag{11}$$

where $\boldsymbol{\eta}(\mathbf{r}(t)) = \mathbf{n}(\mathbf{r}(t)) \times \boldsymbol{\tau}$ and $\boldsymbol{\tau}$ is a random unit vector. Furthermore, as the model nears convergence at this stage, we adopt the SDF-to-density conversion method proposed by TUVR [46], which ensures minimal bias and preserves fine object details. The loss function employed in the second phase is defined as follows:

$$\mathcal{L}_{\text{fine}} = \mathcal{L}_{\text{color}} + \lambda_{\text{eik}}\mathcal{L}_{\text{eik}}(\nabla f) + \lambda_{\text{smooth}}\mathcal{L}_{\text{smooth}}. \tag{12}$$

## 5 Experiments

**Experimental setup.** We carry out experimental evaluations on two benchmark datasets: Tanks and Temples [11] and ScanNet++ [41]. We include several baselines for comparisons: VolSDF [35], NeuralWarp [5], COLMAP [23], NeuS [28], Geo-NeuS [7], Neuralangelo [13] and MonoSDF [43]. We extract mesh through marching cube algorithm with a resolution of 2048 applied across all scenes and report the F-score for suface evaluation. More details are provided in the supplementary materials. Please refer to Appendix H for additional experimental results.

### 5.1 Tanks and Temples

NeuRodin outperforms previous state-of-the-art methods in terms of the average F-score. Owing to our explicit bias correction technique, the barn's roof maintains its structural integrity without

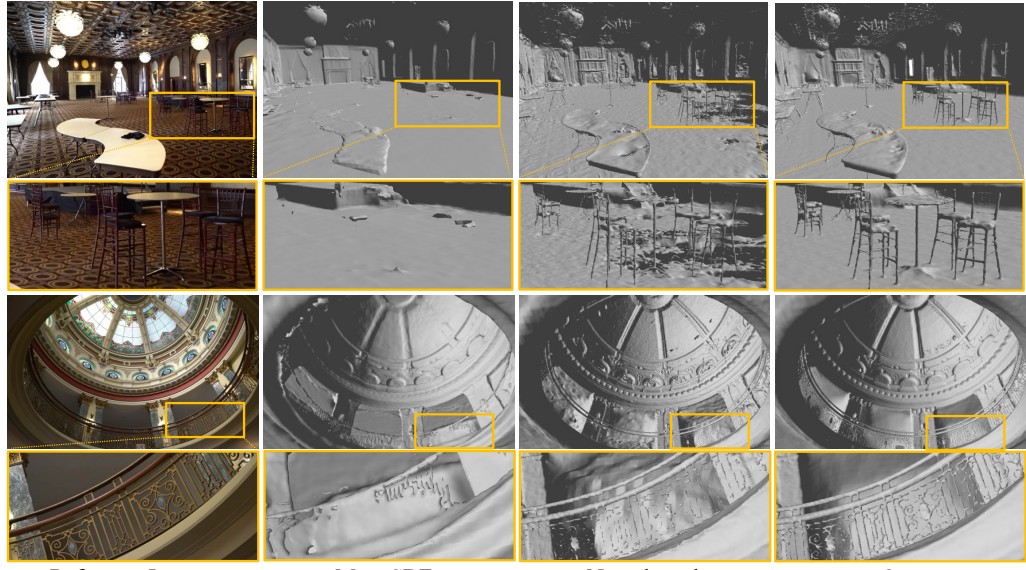

| Reference Image | MonoSDF | Neuralangelo | Ours |

Figure 6: **Quantitative evaluation of our method on the Tanks and Temples advance subset.**

collapsing as shown at the top of Figure 5, a marked improvement over other methods which often fail to prevent such collapse. Thanks to our two-stage optimization approach, we effectively mitigate the issue of excessive geometric regularization as shown at the bottom of Figure 5. Consequently, NeuRodin achieves a more detailed surface representation than Neuralangelo with 1/8 fewer parameters.

It is evident that our method outperforms previous work on the Tanks and Temples advance subset as shown in Table 2. In the comparison in Figure 6, we demonstrate enhanced accuracy and completeness when recon-

| Method | VolSDF | MonoSDF | Neuralangelo* | Neuralangelo | Ours |
|---|---|---|---|---|---|
| Mean | 6.72 | 20.89 | 27.14 | 26.28 | **28.84** |

Table 2: **Quantitative evaluation of our method versus prior work on the Tanks and Temples advance subset.** The **best** performance and the second-best outcomes are highlighted for easy reference.

structing large-scale surfaces, alongside capturing more fine-grained details compared to Neuralangelo. Benefiting from our explicit bias correction in the first stage and the TUVR modeling employed in the second stage, fine structures are restored to the zero level set in the initial phase and maintain a sufficiently small bias in the subsequent phase, thus enabling the refinement of high-quality surfaces.

## 5.2 ScanNet++ Benchmark

Since no public results are available for the ScanNet++ dataset, we randomly selected 8 scenes to construct a benchmark. For more details and results on our ScanNet++ benchmark, please refer to the supplementary materials. Quantitative results are shown in Table 3. We surpassed the methods we compared against in most scenes and achieved comparable results to those with prior knowledge in terms of F-score. We provide more visual result on ScanNet++ dataset in the supplementary.

| F-Score ↑ | Without Prior | | | | | With Prior | |
|---|---|---|---|---|---|---|---|
| method | NeuS | VolSDF | Neuralangelo* | Neuralangelo | Ours | MonoSDF-MLP | MonoSDF-Grid |
| Mean | 0.455 | 0.391 | 0.507 | 0.564 | 0.638 | 0.439 | **0.642** |

Table 3: **Quantitative evaluation of our method versus prior work on the ScanNet++ dataset.** The **best** performance and the second-best outcomes are highlighted for easy reference.

## 5.3 Analysis

**Ablation Study.** To validate the efficacy of the proposed techniques, we performed an ablation study on scene *Meetingroom* from Tanks and Temples dataset. As illustrated in Figure 7 (a), applying a global scale for SDF-to-density conversion results in inaccurate surfaces, primarily due to the

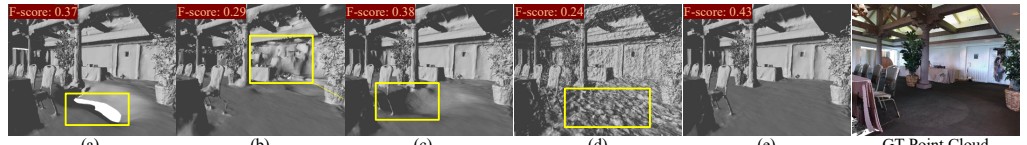

Figure 7: **Ablation results.** (a) Without local scale for SDF-to-density conversion. (b) Without stochastic-step numerical gradient estimation. (c) Without explicit bias correction. (d) Without stage-two refinement. (e) Full model.

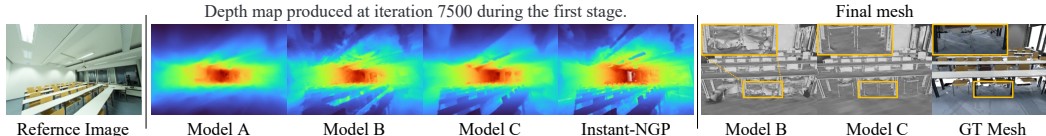

Figure 8: **Analysis on stochastic-step numerical gradient estimation.** We visualize the produced depth maps at iteration 7500 of the first stage. Model A: Change stochastic-step numerical gradient estimation to analytical gradient. Model B: Change stochastic-step numerical gradient estimation to progressive numerical gradient estimation from Neuralangelo. Model C: Ours.

assumption of uniform density across the same level sets. This assumption leads to the convergence of surfaces with subtler textures to incorrect locations. Figure 7 (b) demonstrates that the absence of stochastic-step numerical gradient estimation hinders the model's ability to form arbitrary topologies, leading to incorrect surfaces. The incorrect ground collapsing depicted in Figure 7 (c) is due to the bias in density, creating inconsistencies as shown in Figure 3(c). Without the application of our proposed explicit bias correction, this bias issue causes visibly incorrect surfaces. Figure 7 (d) presents the outcomes of optimization conducted in a single phase; under stochastic-step numerical gradient estimation, the Eikonal loss's preference for smoothness is somewhat compromised, resulting in a rougher surface finish. Finally, Figure 7 (e) showcases our full model, which achieves high-fidelity smooth surfaces while preserving most details. We conducted additional ablation experiments in Appendix H.6.

**Analysis on Stochastic-step Numerical Gradient Estimation.** We further demonstrate experimentally the impact of our stochastic-step numerical gradient estimation on the optimization process, as illustrated in Figure 8. The depth maps produced indicate that Model A is severely constrained by geometric regularization, making it difficult to alter its topological structure during optimization. Model B employs a progressive numerical gradient estimation technique that, even after 7500 steps, does not yield an accurate depth map. However, utilizing our stochastic-step numerical gradient estimation, we achieve an approximately accurate scene geometry in as few as 7500 steps.

At this stage, the depth map of Model C is similar to that of Instant-NGP, yet ours displays a more natural and smooth depth profile. This suggests that our approach, similar to instant-NGP, is capable of freely altering topological shapes for optimization, while also maintaining a natural zero-level set surface.

Furthermore, final mesh of Model B still manifests inaccurate floor collapses, whereas mesh of model C, with our stochastic-step numerical gradient estimation, maintains correct and smooth floors. This is attributed to the fact that the floor, being a large scale, allows our stochasticity to continuously apply the Eikonal constraint on large-scale areas. This results in a natural zero level set in these vast regions. However, for progressive steps, once the step size is reduced beyond a certain point, it no longer imposes the Eikonal constraint on large-scale surfaces, resulting in unnatural zero level sets.

**Analysis on Explicit Bias Correction.** To substantiate the versatility and effectiveness of our explicit bias correction approach, we further conducted experiments to verify its potential as a plug-and-play correction method independent of our full model. We implemented this correction technique on our coarse stage and tested it across various renderers, including NeuS, VolSDF, and TUVR. These experiments were designed to evaluate the adaptability and efficacy of our bias correction in diverse SDF-to-density modelings.

As depicted in Figure 9, the ceiling of the room displays a pronounced bias issue that leads to a collapse. However, with the application of our proposed explicit bias correction approach, the issue of ceiling collapse is significantly ameliorated.

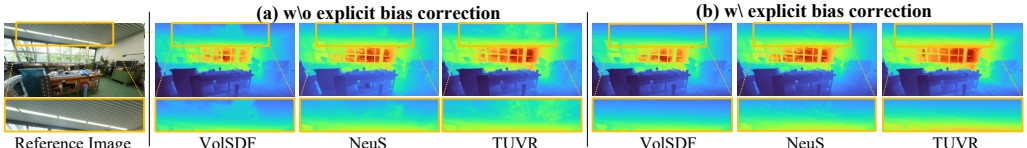

Figure 9: **Analysis on explicit bias correction.** We visualize the produced depth maps with and without our explicit bias correction for different SDF-to-density modelings at the first stage.

## 6 Conclusion

This paper proposes NeuRodin, a two-stage framework for high-fidelity neural surface reconstruction with intricate details. It introduces key designs to tackle SDF-based rendering challenges, notably the local scale adjustment for SDF-to-density conversion, which enables any non-negative value to be achieved, facilitating accurate density derivation from SDF. Additionally, an explicit bias correction method is employed to ensure the geometry of the volume rendering scheme coherently aligns with that of the implicit surface, thereby preventing the emergence of incorrect surfaces. Finally, a two-stage optimization strategy effectively resolves the issue of over-regularization imposed by geometric constraints. Comprehensive experiments demonstrate that NeuRodin simultaneously delivers superior quality.

## Acknowledgments and Disclosure of Funding

This work is funded in part by the National Key R&D Program of China No.2022ZD0160102, and Shanghai Artificial Intelligence Laboratory.

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

# A Analysis of Local Scale in SDF-to-Density Transformation

We further illustrate the importance of the local scale in Figure A, using a simple single plane scenario for explanation. This plane has a low-texture region on the left and a rich-texture region on the right.

Without special constraints, the rendering weight should converge to a Dirac delta function at the surface in the richly textured region and form a scattered distribution in the weakly textured region.

However, under the assumption of a global scale factor, all areas on the plane follow the same distribution which will be derived in the following sections. This means that both richly textured and weakly textured regions share the same density bias, preventing the surface from converging correctly to the richly textured surface with higher certainty.

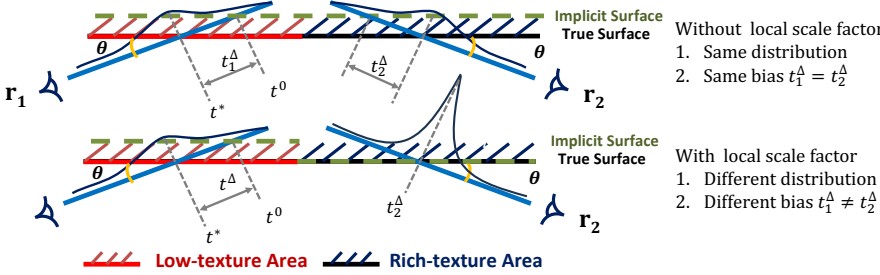

Figure 10: **Explanation of the motivation behind the use of the local scale factor.**

By introducing the local scale factor, the most significant difference is that the distribution of rendering weights along the ray is no longer uniform. The network can adaptively converge in the richly textured regions, and the density bias in these areas is no longer affected by the low-texture regions.

In Adaptive shells, their focus is primarily on rendering quality and they did not evaluate surface reconstruction metrics. A straightforward application to surface reconstruction can lead to issues such as increased density bias (as $w(t)$ is also a function of the scale factor $s(t)$). We tackle this by implementing special designs to ensure effectiveness, such as gradually scheduling the lower bound of the scale factor to correct relatively small biases and the explicit bias correction.

# B Analysis of Density Bias

In this section, we analyze the density bias and discuss the main disadvantage of previous work (NeuS [28], VolSDF [35] and TUVR [46]). According to Equation (8), the rendering weight maximum point $t^*$ should satisfy

$$t^* = \arg\max_{t \in (0, +\infty)} T(t)\sigma(\mathbf{r}(t)). \tag{13}$$

The derivation of rendering weight $w(t) = T(t)\sigma(\mathbf{r}(t))$ respected to $t^*$ should equal to zero:

$$
\begin{aligned}
\left.\frac{\partial w(t)}{\partial t}\right|_{t=t^*} &= \left.\frac{\partial(T(t)\sigma(\mathbf{r}(t)))}{\partial t}\right|_{t=t^*} \\
&= \left.\frac{\partial T(t)}{\partial t}\right|_{t=t^*}\sigma(\mathbf{r}(t^*)) + \left.\frac{\partial\sigma(\mathbf{r}(t))}{\partial t}\right|_{t=t^*}T(t^*) \\
&= \left(\sigma^2(\mathbf{r}(t^*)) - \left.\frac{\partial\sigma(\mathbf{r}(t))}{\partial t}\right|_{t=t^*}\right)\exp\left(-\int_0^{t^*}\sigma(u)du\right) \\
&= 0.
\end{aligned}
\tag{14}
$$

Then, we have

$$\sigma^2(\mathbf{r}(t^*)) = \left.\frac{\partial\sigma(\mathbf{r}(t))}{\partial t}\right|_{t=t^*}. \tag{15}$$

NeuS [28] and TUVR [46] seek a modeling approach for $\sigma(\mathbf{r}(t))$ such that it also satisfies the above equation at the point where the SDF value is zero $f(r(t^0)) = 0$. NeuS only satisfies this under the

first-order approximation of the distribution of the SDF along the ray, while TUVR extends this to arbitrary distributions.

Although TUVR's modeling ensures that the derivative of the rendering weight is zero at the point $t^0$ where the SDF value is zero, this does not mean that $t^0$ is the location of the global maximum of the weight.

During the optimization process, the distribution of rendering weight along the ray is a complex non-convex function. Therefore, TUVR only guarantees that $t^0$ is at a local maxima. Therefore, bias persists throughout the optimization process. We have shown more analysis on TUVR in Section D.

Here we take VolSDF's SDF-to-density modeling (3) as an example. And considering the simplest case, where the ray intersects with a single plane. In other words, the implicit surface at the current iteration is a single plane. In this case, if the angle between the ray and the plane is $\theta$, and the distance from the ray's origin to the plane is $d$, then the distribution of the SDF along the ray can be expressed as $f(\mathbf{r}(t)) = -t\sin\theta + d$. In Equation (15), the expression on the right-hand side is

$$\frac{\partial\sigma(\mathbf{r}(t))}{\partial t} = -\frac{1}{2s^2}\frac{\partial f(\mathbf{r}(t))}{\partial t}\exp\left(\frac{|f(\mathbf{r}(t))|}{s}\right). \tag{16}$$

And the left-hand side is

$$\sigma^2(\mathbf{r}(t)) = \begin{cases} \frac{1}{4s^2}\exp\left(\frac{-2f(\mathbf{r}(t))}{s}\right) & \text{if } f(\mathbf{r}(t)) \geq 0, \\ \frac{1}{s^2}\left(1 - \frac{1}{2}\exp\left(\frac{f(\mathbf{r}(t))}{s}\right)\right)^2 & \text{if } f(\mathbf{r}(t)) < 0. \end{cases} \tag{17}$$

If $f(t^*) \geq 0$, then we have

$$\frac{1}{4s^2}\exp\left(\frac{-2f(\mathbf{r}(t^*))}{s}\right) = -\frac{1}{2s^2}\frac{\partial f(\mathbf{r}(t))}{\partial t}\bigg|_{t=t^*}\exp\left(\frac{-f(\mathbf{r}(t^*))}{s}\right)$$

$$\exp\left(\frac{-f(\mathbf{r}(t^*))}{s}\right) = -2\frac{\partial f(\mathbf{r}(t))}{\partial t}\bigg|_{t=t^*} \tag{18}$$

$$\exp\left(\frac{t^*\sin\theta - d}{s}\right) = 2\sin\theta.$$

We can directly obtain the closed-form solution for $t^*$ only if $\sin\theta \leq 0.5$

$$t^* = \frac{s\ln(2\sin\theta) + d}{\sin\theta}. \tag{19}$$

If $f(t^*) < 0$, then we have

$$\exp\left(\frac{-t\sin\theta + d}{s}\right)\sin\theta = 2\left(1 - \frac{1}{2}\exp\left(\frac{-t\sin\theta + d}{s}\right)\right)^2 \tag{20}$$

Let $\exp\left((-t\sin\theta + d)/s\right)$ be denoted as $m$. It's easy to see that the above equation has a solution $m^* = 2 + \sin\theta - \sqrt{\sin^2\theta + 4\sin\theta}$ which can be obtained using the quadratic formula only if $\sin\theta > 0.5$, therefore

$$t^* = \frac{s\ln(1/m^*) + d}{\sin\theta}. \tag{21}$$

After organizing the two situations, can obtain the results:

$$t^* = \frac{s\ln(k) + d}{\sin\theta}, \quad k = \begin{cases} 2\sin\theta & \text{if } \sin\theta \leq 0.5, \\ 1/m^* & \text{if } \sin\theta > 0.5. \end{cases} \tag{22}$$

In this case, the closed-form solution for $t^0$ can be directly obtained as $t^0 = d/\sin\theta$. Then the distance between $t^*$ and $t^0$ is

$$t^\Delta = t^* - t^0 = \frac{\ln k}{\sin\theta}s. \tag{23}$$

It can be observed that $t^\Delta$ is a linear function of the scale factor $s$. If the scale factor $s$ is relatively large, the implicit surface is far from the regions that contribute most to the color. At that time, the geometry regularization on the implicit surface acts on the incorrect surface, resulting in visible defects in the final mesh. Our explicit bias correction aims to minimize $t^\Delta$ during the coarse stage, so that the geometric regularization can be applied to the correct surface.

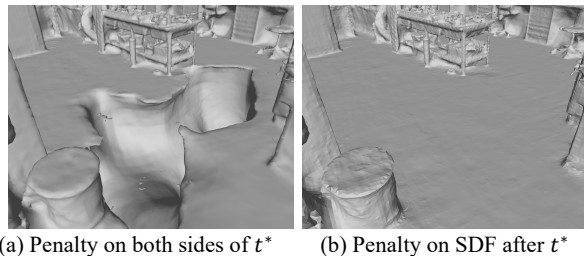

| (a) Penalty on both sides of $t^*$ | (b) Penalty on SDF after $t^*$ |

Figure 11: **Visual results on ScanNet++ with different designs of our explicit bias correction.** Penalty on both sides of $t^*$ will lead to erroneous surface

## C Design of Explicit Bias Correction

Although aligning $t^*$ with the SDF zero-crossing is intuitive, our experiments showed that this approach requires special design considerations. We tried simply constraining $f(t^*)$ to approach zero but found that visible surface defects persisted. We also attempted to constrain the SDF values both before and after $t^*$, which led to very strange optimized surfaces, likely due to overly strong constraints on the SDF. We have shown the visual result in Figure 11.

The key to our method's design lies in correcting large relative bias with explicit loss functions, while smaller biases are corrected by gradually scheduling the lower bound of the scale factor. We empirically found that visible surface errors are always caused by $t^*$ is being before SDF zero-crossing points $t^0$. Therefore, we penalize the SDF value just after $t^*$ to trend towards negative values, which prevents $t^*$ from being before $t^0$. For the case where $t^*$ is after $t^0$, we can directly address it by gradually scheduling the lower bound of the scale factor, as the bias in this situation is relatively small.

We provided a mathematical explanation under the assumption of a sufficiently small local surface. According to Section B the distance between $t^*$ and $t^0$ is as Equation (23).

When $t^\Delta < 0$, $t^*$ is before $t^0$, and when $t^\Delta > 0$, $t^*$ is after $t^0$. We visualized the values of $t^\Delta$ under different $\theta$ in Figure 12. We found that when $t^*$ is before $t^0$ ($\sin \theta$ less than 0.5), the relative bias is significantly greater than when $t^*$ is after $t^0$. Therefore, we penalize the SDF value just after $t^*$ to prevent $t^*$ being before $t^0$.

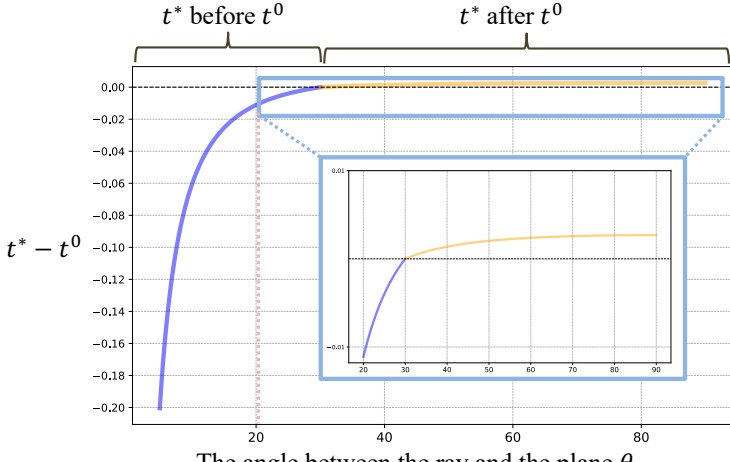

Figure 12: **The distance between $t^*$ and $t^0$ as the angle between the ray and the single plane varies.** We find that when $t^*$ precedes $t^0$, a significantly larger relative bias is observed.

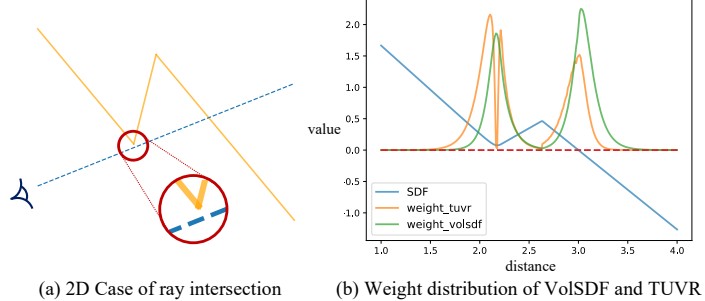

(a) 2D Case of ray intersection      (b) Weight distribution of VolSDF and TUVR

Figure 13: **A particular scenario for density bias.** In this situation TUVR exhibits a more pronounced bias compared to VolSDF.

## D    Analysis of TUVR

Previously, we mentioned that TUVR only proves $t^0$ as a local maximum for rendering weights, not a global one. Here, we present a scenario where bias exists in the modeling of TUVR. Consider a wall composed of three planes in space, with light passing through it, as shown in Figure 13 (a). In this situation, the rendering weights, as illustrated in Figure 13 (b), exhibit a bias in TUVR; although TUVR ensures a local peak at the SDF zero crossing point, the rendering weight is greater at a previous location. In this scenario, the bias in TUVR may even be greater than that in VolSDF.

Next, we analyze the unbiasedness of TUVR under the assumption of a local scale factor. To simplify the equations, we will henceforth abbreviate all $\mathbf{r}(t)$ as $t$ and all $\frac{\partial \cdot}{\partial t}$ as $\cdot'(t)$. When using a local scale factor, the density represented under TUVR modeling is:

$$\sigma(t) = \begin{cases} \frac{1}{s(t)} \exp\left(\frac{-f(t)}{s(t)|f'(t)|}\right) & \text{if } f(t) \geq 0, \\ \frac{2}{s(t)}\left(1 - \frac{1}{2}\exp\left(\frac{f(t)}{s(t)|f'(t)|}\right)\right) & \text{if } f(t) < 0. \end{cases} \tag{24}$$

When $f(t) \geq 0$, the left-hand side of Equation (15) is:

$$\sigma'(t) = -\frac{s'(t)}{s^2(t)}\exp\left(-\frac{f(t)}{s(t)|f'(t)|}\right) + \frac{1}{s(t)}\left(-\frac{\left|\frac{f(t)}{|f'(t)|}\right|' s(t) - \frac{f(t)}{|f'(t)|}s'(t)}{s^2(t)}\right)\exp\left(-\frac{f(t)}{s(t)|f'(t)|}\right) \tag{25}$$

When $f(t) = 0$, the above expression can be simplified to:

$$\sigma'(t) = -\frac{s'(t)}{s^2(t)} - \frac{f'(t)}{s^2(t)|f'(t)|} \tag{26}$$

And we only consider the scenario where the light ray enters the plane, namely $f'(t) < 0$. So we have

$$\sigma'(t) = \frac{-s'(t) + 1}{s^2(t)} \tag{27}$$

At this point, $\sigma^2(t) = 1/s^2(t)$, so Equation (15) is only satisfied when $s'(t) = 0$, meaning the scale factor is a constant. Therefore, under the assumption of a local scale factor, TUVR's local unbiasedness (ensuring the SDF zero crossing point is a local maximum in rendering weights) cannot be achieved. When $f(t) < 0$, we can also arrive at a similar conclusion.

## E    Explanation of Stochastic Gradients

Our stochastic gradient estimation introduces some uncertainty into the true normals. For large-scale features, the Eikonal loss with varying step sizes can still be successfully minimized (since SDF near

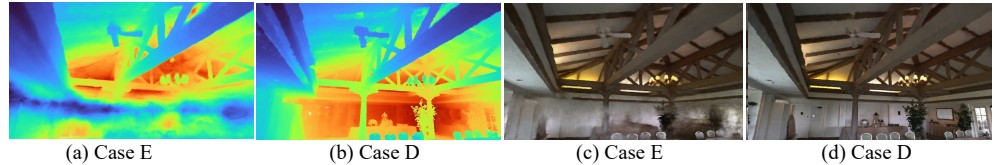

| (a) Case E | (b) Case D | (c) Case E | (d) Case D |

Figure 14: **Visual results on Tanks and Temples from iteration 25,000 at the first stage.**

a large-scale plane should satisfy the Eikonal equation for different step sizes). In other words, the variance of stochastic gradients is small for large-scale features, making it easier for the model to minimize the Eikonal loss. However, for fine details, the random step sizes lead to high variance in the estimated normals, which reduces the impact of the Eikonal loss and makes it more challenging for the model to minimize samples with high variance.

## F    Impact of Color Conditioning on Normal

In Figure 14, we experimentally observed that in indoor scenes, when color conditioning on normals is applied, the optimization process becomes very slow and adversely affects the optimization results (both surface and rendering quality). However, this issue is nearly absent in outdoor scenes. We believe this is primarily due to the convergence behavior of the scale factor. Indoor scenes often have many weakly textured areas, leading to larger scale factors and a greater scatter distribution. Additionally, color conditioning on normals, which is intended for geometry disentanglement as mentioned in IDR and resonable for surface points, results in many details being optimized to incorrect positions before convergence is achieved.

However, our use of stochastic normals helps mitigate these erroneous surfaces. For detailed regions, the estimated normals have greater variance, which alleviates the impact of incorrect surfaces. This is also demonstrated in our ablation experiments, as shown in the Table 4. Cases D and E are as follows: Case D: Excludes the

| F-score ↑ | D | E | Full Model |
|---|---|---|---|
| Courthouse (outdoor) | 0.11 | 0.11 | 0.21 |
| Meetingroom (indoor) | 0.35 | 0.21 | 0.43 |

Table 4: **Impact of color conditioning on normal in different scenes.**

stage 1 Eikonal loss but incorporates color conditioning based on the estimated normal. Case E: Excludes the stage 1 Eikonal loss but includes color conditioning based on the analytical normal.

## G    Experimental Details

### G.1    Datasets

We carry out experimental evaluations on two benchmark datasets: Tanks and Temples [11] and ScanNet++ [41]. The Tanks and Temples dataset is characterized by its large-scale, diverse real-world scenes, both indoors and outdoors. For our experiments, we utilize six scenes from the training subset, consistent with the scenes employed in Neuralangelo, to maintain comparability. Additionally, we extend our validation to four expansive indoor scenes from the advanced subset to further assess the robustness of our method. Turning to the ScanNet++ dataset, it is distinguished by its high-quality indoor scenes, supplemented with DSLR-quality images. From this dataset, we have selected eight scenes for our analysis.

### G.2    Baselines

For the Tanks and Temples dataset, our methodology is compared against several prominent methods, including: NeuralWarp [5], COLMAP [23], NeuS [28], Geo-NeuS [7], and Neuralangelo [13]. In the context of the ScanNet++ dataset, our approach is contrasted with methods lacking prior knowledge, such as VolSDF [35], NeuS [28], and Neuralangelo [13]. Additionally, we evaluate our approach against methods incorporating pretrained prior information, notably MonoSDF [43]. It should be noted that our efforts to reproduce Neuralangelo for indoor scenes were unsuccessful. Instead, we employed the implementation of Bakedangelo from [42], which serves as an enhanced version of Neuralangelo. Bakedangelo utilizes the same proposal network as our setup. We discovered that manually adjusting the global scale of SDF-to-density conversion in Bakedangelo significantly

| Scene | Metric | NeuS | VolSDF | Neuralangleo* | Neuralangelo | Ours | MonoSDF-MLP | MonoSDF-Grid |
|---|---|---|---|---|---|---|---|---|
| 0e75f3c4d9 | Acc | 0.053 | 0.053 | 0.326 | 0.121 | 0.166 | 0.041 | 0.064 |
| | Comp | 0.065 | 0.078 | 0.039 | 0.053 | 0.029 | 0.038 | 0.027 |
| | Prec | 0.433 | 0.395 | 0.381 | 0.525 | 0.584 | 0.611 | 0.574 |
| | Recal | 0.437 | 0.363 | 0.734 | 0.719 | 0.790 | 0.665 | 0.705 |
| | F-score | 0.435 | 0.378 | 0.502 | 0.607 | 0.671 | 0.637 | 0.633 |
| 036bce3393 | Acc | 0.050 | 0.053 | 0.184 | 0.168 | 0.028 | 0.062 | 0.037 |
| | Comp | 0.066 | 0.098 | 0.028 | 0.023 | 0.029 | 0.097 | 0.037 |
| | Prec | 0.507 | 0.475 | 0.509 | 0.520 | 0.688 | 0.383 | 0.606 |
| | Recal | 0.492 | 0.372 | 0.754 | 0.780 | 0.733 | 0.333 | 0.656 |
| | F-score | 0.499 | 0.417 | 0.608 | 0.624 | 0.710 | 0.356 | 0.630 |
| 108ec0b806 | Acc | 0.047 | 0.049 | 0.121 | 0.092 | 0.044 | 0.059 | 0.041 |
| | Comp | 0.088 | 0.097 | 0.060 | 0.050 | 0.053 | 0.113 | 0.045 |
| | Prec | 0.519 | 0.475 | 0.389 | 0.498 | 0.597 | 0.395 | 0.575 |
| | Recal | 0.432 | 0.372 | 0.542 | 0.624 | 0.587 | 0.303 | 0.576 |
| | F-score | 0.472 | 0.417 | 0.453 | 0.554 | 0.592 | 0.343 | 0.576 |
| 21d970d8de | Acc | 0.046 | 0.070 | 0.247 | 0.261 | 0.075 | 0.054 | 0.042 |
| | Comp | 0.050 | 0.060 | 0.071 | 0.049 | 0.031 | 0.062 | 0.031 |
| | Prec | 0.526 | 0.383 | 0.403 | 0.403 | 0.588 | 0.365 | 0.580 |
| | Recal | 0.565 | 0.410 | 0.595 | 0.656 | 0.726 | 0.371 | 0.676 |
| | F-score | 0.545 | 0.396 | 0.481 | 0.499 | 0.650 | 0.368 | 0.624 |
| 355e5e32db | Acc | 0.042 | 0.043 | 0.073 | 0.079 | 0.034 | 0.043 | 0.032 |
| | Comp | 0.075 | 0.071 | 0.038 | 0.034 | 0.038 | 0.060 | 0.031 |
| | Prec | 0.534 | 0.501 | 0.575 | 0.575 | 0.672 | 0.439 | 0.657 |
| | Recal | 0.465 | 0.447 | 0.701 | 0.730 | 0.689 | 0.404 | 0.683 |
| | F-score | 0.497 | 0.475 | 0.632 | 0.643 | 0.681 | 0.421 | 0.669 |
| 578511c8a9 | Acc | 0.094 | 0.080 | 0.254 | 0.222 | 0.103 | 0.063 | 0.039 |
| | Comp | 0.174 | 0.212 | 0.057 | 0.030 | 0.044 | 0.171 | 0.044 |
| | Prec | 0.373 | 0.354 | 0.378 | 0.465 | 0.520 | 0.322 | 0.608 |
| | Recal | 0.328 | 0.271 | 0.599 | 0.737 | 0.623 | 0.264 | 0.657 |
| | F-score | 0.349 | 0.307 | 0.463 | 0.570 | 0.567 | 0.290 | 0.631 |
| 7f4d173c9c | Acc | 0.080 | 0.097 | 0.414 | 0.420 | 0.155 | 0.094 | 0.138 |
| | Comp | 0.055 | 0.075 | 0.026 | 0.024 | 0.028 | 0.030 | 0.022 |
| | Prec | 0.503 | 0.457 | 0.437 | 0.457 | 0.657 | 0.704 | 0.708 |
| | Recal | 0.496 | 0.436 | 0.771 | 0.799 | 0.732 | 0.713 | 0.813 |
| | F-score | 0.500 | 0.446 | 0.557 | 0.582 | 0.693 | 0.709 | 0.757 |
| 09c1414f1b | Acc | 0.170 | 0.085 | 0.348 | 0.106 | 0.081 | 0.070 | 0.050 |
| | Comp | 0.116 | 0.285 | 0.076 | 0.242 | 0.056 | 0.105 | 0.080 |
| | Prec | 0.348 | 0.347 | 0.289 | 0.475 | 0.545 | 0.418 | 0.651 |
| | Recal | 0.331 | 0.256 | 0.470 | 0.403 | 0.530 | 0.363 | 0.589 |
| | F-score | 0.339 | 0.294 | 0.358 | 0.436 | 0.537 | 0.389 | 0.618 |
| Mean | Acc | 0.073 | 0.066 | 0.246 | 0.184 | 0.086 | 0.061 | **0.055** |
| | Comp | 0.086 | 0.122 | 0.049 | 0.063 | **0.039** | 0.085 | 0.040 |
| | Prec | 0.468 | 0.423 | 0.420 | 0.490 | 0.606 | 0.455 | **0.620** |
| | Recal | 0.443 | 0.366 | 0.646 | **0.681** | 0.676 | 0.427 | 0.669 |
| | F-score | 0.455 | 0.391 | 0.507 | 0.564 | 0.638 | 0.439 | **0.642** |

Table 5: **ScanNet++ Benchmark** We established benchmarks for eight different scenes within the ScanNet++ dataset.

mitigates the optimization issues encountered with Neuralangelo in indoor scenes. Additionally, we eliminated the unnecessary 360 unbound setting and background modeling for indoor scenes. Therefore, in the context of the Tanks and Temples advanced subset and ScanNet++, we report the results obtained with Bakedangelo as a substitute for those of Neuralangelo.

### G.3 Evaluation Metrics

Mesh is extracted through marching cube algorithm with a resolution of 2048 applied across all scenes. For the Tanks and Temples dataset, the evaluation metrics on the training subset are computed using the official Python script provided by the dataset's maintainers [1]. Meanwhile, for the four scenes from the advanced subset, our reconstructed results are submitted to the evaluation server [2], which calculates the evaluation metrics. For the ScanNet++ dataset, we calculate the F-score with a threshold of 0.025 to compare the resultant meshes with the ground truth mesh, which is derived from the point cloud captured by a laser scanner.

---

[1]https://github.com/isl-org/TanksAndTemples/tree/master/python toolbox/evaluation
[2]https://www.tanksandtemples.org/

## G.4 Implementation Details

NeuRodin utilizes a multi-resolution hash grid for encoding, spanning from $2^5$ to $2^{11}$ across 16 levels. Each hash entry possesses a channel size of 2 for room-level scenes such as ScanNet++, which is adjusted to 8 larger-scale scenes like Tanks and Temples. The maximum number of hash entries for each resolution is set at $2^{19}$. We incorporate per-image appearance encoding in the style of NeRF-W [15] while employing a proposal network [1] based on a compact hash grid. For the outdoor scene, we model the background using an additional network [45] with the hash grid. For the ScanNet++ datasets, we sample 512 pixels per iteration. In the case of the Tanks and Temples dataset, we sample 1024 pixels during the first stage and escalate to 8192 pixels for the second stage. We set the weights $\lambda_{\text{eik}}$ and $\lambda_{\text{smooth}}$ to be 0.01 and 0.005, respectively. For outdoor scenes, we set $\lambda_{\text{bias}}$ to 0.1. For indoor scenes, we progressively increase $\lambda_{\text{bias}}$ from 0.001 to 0.05 over the first 10,000 iterations through an exponential adjustment. Bakedangelo samples 8192 pixels per iteration in large indoor scenes of Tanks and Temples, aligning with the settings employed by Neuralangelo. In the context of room-level scenarios within ScanNet++, the batch size is adjusted to 1024 pixels per iteration. Our implementation of the method employs PyTorch [20] and utilizes the Adam optimizer, with a learning rate of 0.001 applied to both the hash grid and the network, alongside a weight decay set at 0.01. For the background model, we set the learning rate to 0.01. The total training steps amount to 300k, with the learning rate for the foreground model being decayed by a factor of 10 at 160k and 240k steps. For the background model, we employ an exponential schedule for the learning rate, reducing it to 0.0001. For the proposal network, the learning rate is decayed by a factor of 3 at steps 150k, 225k, and 270k. All our experiments were conducted on an A100 40G GPU. We roughly require 7 GPU hours to complete the reconstruction of an indoor scene from the ScanNet++ dataset. For large-scale scenes, the reconstruction takes approximately 18 GPU hours.

## G.5 Implementation of the Explicit Bias Correction

In the actual implementation of explicit bias correction, when inferring the SDF at $\epsilon_{\text{bias}}$ after the point $\mathbf{r}(t^*)$ along the ray in Equation (8), we also infer the SDF at $\epsilon_{\text{bias mask}}$ beyond $\mathbf{r}(t^*)$ along the ray as a mask. If $f(\mathbf{r}(t^* + \epsilon_{\text{bias mask}}))$ is less than zero, we do not apply the bias loss to this particular ray. We have found that this approach effectively prevents incorrect alignments that may be caused by our approximate estimation of $t^*$. For outdoor scenes, we simply determine whether a ray is cast towards the background by checking if there exists a negative value of SDF at any sampled point along the ray. If so, we similarly refrain from performing bias correction. We have configured $\epsilon_{\text{bias mask}}$ to 0.001 for large-scale scenes, such as the Tanks and Temples dataset. For room-level scenes, such as the ScanNet++ dataset, we have set it to 0.01.

## G.6 Implementation Details of the Two-Stage Optimization

The local scale modeling in Equation (5) can impede the model's convergence to the surface to some extent. Therefore, we manually adjust the lower bound $s_{\text{coarse}}$ for the scale to prevent the ambiguity that arises from excessively small scales. Analogous to the manual adjustments made in the first phase, our objective in this stage is to facilitate convergence from volume rendering to surface

| Metric | F-Score (%) ↑ | | | | |
|--------|--------|---------|---------------|--------------|------|
| Scene | VolSDF | MonoSDF | Neuralangelo* | Neuralangelo | Ours |
| Auditorium | 3.16 | 10.97 | 14.32 | 14.09 | **16.03** |
| Ballroom | 11.61 | 29.30 | 32.21 | 28.93 | **33.10** |
| Courtroom | 7.71 | 21.58 | **36.53** | 32.81 | 33.53 |
| Museum | 4.41 | 21.71 | 25.49 | 29.28 | **32.71** |
| Mean | 6.72 | 20.89 | 27.14 | 26.28 | **28.84** |

Table 6: **Quantitative evaluation of our method versus prior work on the Tanks and Temples advance subset.** The **best** performance and the second-best outcomes are highlighted for easy reference.

rendering, thereby aligning the implicit surface with volume rendering completely. To achieve this, we exponentially increase the lower bound of the scale $s_{\text{fine}}$ to a substantial value. In all experiments, we set the value of $s_{\text{coarse}}$ to 100 and $s_{\text{fine}}$ to 3000.

| | | 24 | 37 | 40 | 55 | 63 | 65 | 69 | 83 | 97 | 105 | 106 | 110 | 114 | 118 | 122 | Mean |
|---|---|---|---|---|---|---|---|---|---|---|---|---|---|---|---|---|---|
| → | NeRF | 1.90 | 1.60 | 1.85 | 0.58 | 2.28 | 1.27 | 1.47 | 1.67 | 2.05 | 1.07 | 0.88 | 2.53 | 1.06 | 1.15 | 0.96 | 1.49 |
| | VolSDF | 1.14 | 1.26 | 0.81 | 0.49 | 1.25 | 0.70 | 0.72 | 1.29 | 1.18 | 0.70 | 0.66 | 1.08 | 0.42 | 0.61 | 0.55 | 0.86 |
| | NeuS | 1.00 | 1.37 | 0.93 | 0.43 | 1.10 | 0.65 | 0.57 | 1.48 | 1.09 | 0.83 | 0.52 | 1.20 | 0.35 | 0.49 | 0.54 | 0.84 |
| CD (mm) | Neuralangelo | 0.37 | 0.72 | 0.35 | 0.35 | 0.87 | 0.54 | 0.53 | 1.29 | 0.97 | 0.73 | 0.47 | 0.74 | 0.32 | 0.41 | 0.43 | 0.61 |
| | Ours | 0.37 | 0.65 | 0.31 | 0.36 | 0.93 | 0.54 | 0.63 | 1.30 | 1.07 | 0.68 | 0.52 | 0.61 | 0.32 | 0.41 | 0.37 | 0.60 |

Table 7: **Quantitative results on DTU Benchmark.**

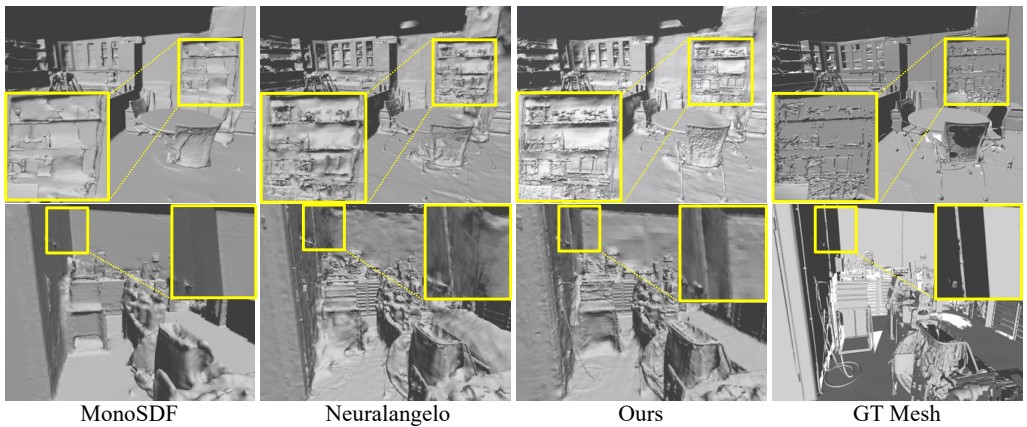

|  MonoSDF  |  Neuralangelo  |  Ours  |  GT Mesh  |

Figure 15: **Quantitative evaluation of our method on the ScanNet++ dataset.**

# H  More Experimental Results

## H.1  Experimental Results on the Scannet++ Benchmark

For every scene, we use high-quality DSLR camera images from all frames for our experiments. For methods that do not use prior knowledge, we downsample the images from $1752 \times 1168$ to $876 \times 584$ for training. For methods that do use prior knowledge, we do something similar to MonoSDF. We first crop the images to $1152 \times 1152$, then downsample them to $384 \times 384$ before feeding them into the Omnidata model [6] to predict geometry cues. First, we scale the pose to be centered within a bounding sphere of radius equal to 1. Subsequently, we scale the camera pose by 0.8 to ensure that all scene boundaries are contained within the bounding sphere. We apply the marching cubes algorithm to a cube with dimensions -1 to 1 at a resolution of 2048. Then, we evaluate the metrics: Accuracy (Acc), Completeness (Comp), Precision (Pred), Recall (Rec), and F-score. The final result is shown in Table 5. Here, we also present some visual results in Figure 15.

## H.2  Experimental Results on the DTU Benckmark

Although our design is not specifically tailored for single-object datasets, we validated our method on the DTU Benchmark. The results are shown in the Table 7. We found that, even without special parameter tuning, our method achieves results comparable to Neuralangelo and surpasses other baseline methods.

## H.3  Experimental Results on the Tanks and Temples Advance Subset

In this section, we present the individual results for each scene within the advanced subset of the Tanks and Temples dataset, as shown in Table 6. Except for the Courtroom scene, our method significantly outperforms comparative approaches in the other three large-scale indoor scenes, demonstrating the effectiveness of our approach. We present a portion of the mesh for the museum data in Figure 16 and more results in Figure 17, highlighting our method's significant capability in capturing details.

## H.4  Comparison with TUVR

Since TUVR is not open-source and only reports metrics on the DTU dataset, we reproduced its unbiased SDF-to-density technique and combined it with a hash grid to create the TUVR-Grid method for comparison. Additionally, we compared our results on the DTU dataset with the reported results in the paper that did not use MVS priors (TUVR-MLP). The results are shown in the table below.

The performance of TUVR is not ideal on all datasets because its unbiased nature is not fully guaranteed, and it is also somewhat affected by over-regularization.

| F-score ↑ | TUVR-Grid | Ours |
|---|---|---|
| Barn | 0.57 | 0.70 |
| Caterpillar | 0.25 | 0.36 |
| Courthouse | 0.11 | 0.21 |
| Meetingroom | 0.03 | 0.43 |
| Truck | 0.33 | 0.47 |
| Ignatius | 0.66 | 0.86 |
| Mean | 0.40 | 0.51 |

Table 8: **Comparison with TUVR on the Tanks and Temples training set.**

| | | 24 | 37 | 40 | 55 | 63 | 65 | 69 | 83 | 97 | 105 | 106 | 110 | 114 | 118 | 122 | Mean |
|---|---|---|---|---|---|---|---|---|---|---|---|---|---|---|---|---|---|
| CD → | TUVR-MLP | 0.72 | 0.77 | 0.67 | 0.37 | 0.93 | 0.58 | 0.61 | 1.23 | 1.15 | 0.65 | 0.56 | 1.08 | 0.34 | 0.45 | 0.47 | 0.71 |
| | TUVR-Grid | 0.84 | 0.81 | 1.44 | 0.37 | 1.23 | 0.69 | 0.78 | 1.16 | 1.23 | 0.65 | 0.54 | 1.32 | 0.34 | 0.43 | 0.54 | 0.82 |
| | Ours | 0.37 | 0.65 | 0.31 | 0.36 | 0.93 | 0.54 | 0.63 | 1.30 | 1.07 | 0.68 | 0.52 | 0.61 | 0.32 | 0.41 | 0.37 | 0.60 |

Table 9: **Comparison with TUVR on the DTU Benchmark.**

## H.5  Image Reconstruction Comparison

While our design is primarily intended for surface reconstruction tasks, we have also compared the results of NeuS and Neuralangelo using the image quality evaluation methods from Neuralangelo, across different parameter scales.

As shown in Table 10, with fewer parameters, our method (Ours-19) achieves image reconstruction quality close to that of Neuralangelo. With a larger number of parameters, our method (Ours-22) surpasses Neuralangelo in terms of image reconstruction quality.

| PSNR | NeuS | Neuralangelo-22 | Ours-19 | Ours-22 |
|---|---|---|---|---|
| Mean | 24.58 | 27.24 | 26.90 | 27.67 |

Table 10: **Image Reconstruction Results on the Tanks and Temples training set.**

## H.6  More Ablation Study

We conducted additional ablation experiments on the Tanks and Temples training set and we included two more cases to verify the role of Eikonal loss in maintaining the natural zero level set during the first stage, as well as the impact of color condition on normals.

The five cases are: A: Without the local scale factor. B: Changing the stage 1 estimated gradient to the analytical gradient. C: Without explicit bias correction. D: Without stage 1 Eikonal loss, but with color conditioning on the estimated normal. E: Without stage 1 Eikonal loss, but with color conditioning on the analytical normal.

| F-score ↑ | A | B | C | D | E | Full Model |
|---|---|---|---|---|---|---|
| Barn | 0.70 | 0.58 | 0.68 | 0.54 | 0.56 | 0.70 |
| Caterpillar | 0.33 | 0.33 | 0.36 | 0.32 | 0.33 | 0.36 |
| Courthouse | 0.24 | 0.12 | 0.19 | 0.11 | 0.11 | 0.21 |
| Meetingroom | 0.37 | 0.29 | 0.38 | 0.35 | 0.21 | 0.43 |
| Truck | 0.46 | 0.39 | 0.47 | 0.42 | 0.43 | 0.47 |
| Ignatius | 0.81 | 0.79 | 0.86 | 0.77 | 0.78 | 0.86 |
| Mean | 0.49 | 0.42 | 0.49 | 0.42 | 0.40 | 0.51 |

Table 11: **More ablation study on Tanks and Temples training set.**

The quantitive results are in Table 11. The results from cases A, B, and C demonstrate the effectiveness of the techniques we proposed. The results from cases D and E indicate the necessity of applying Eikonal loss during the first stage.

## I  Limitation

Though NeuRodin has capabilities in reconstruction, it falls short in certain areas. Specifically, it struggles to faithfully reconstruct the correct surface in areas that are textureless and less observed. NeuRodin is also incapable of handling situations with strong ambiguity. Additionally, density is not guaranteed to be unbiased, hence, bias will always exist though the SDF-to-density conversion.

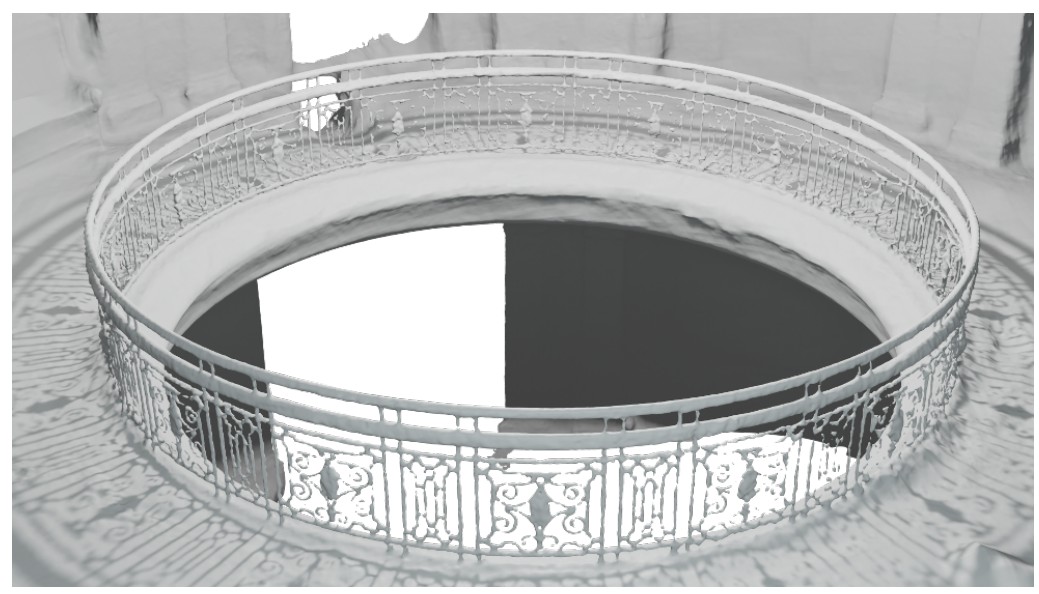

Figure 16: **Recovered mesh of the *Museum* data from the Tanks and Temples dataset.**

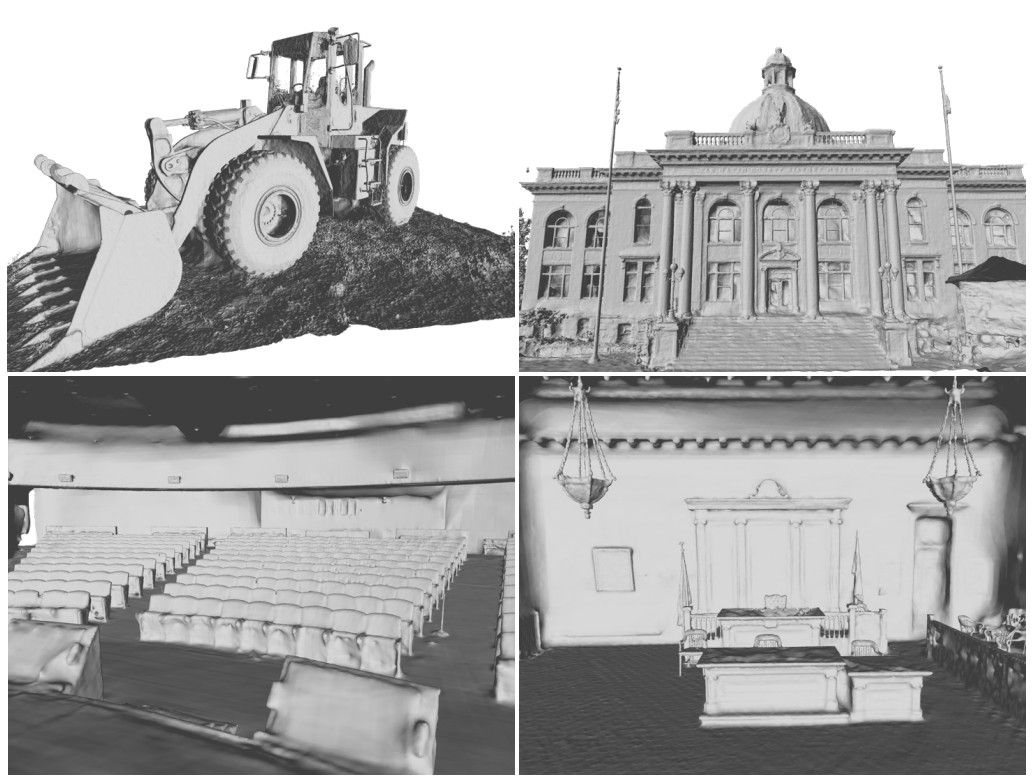

Figure 17: **More results from the Tanks and Temples dataset.**

Lastly, the time taken for the reconstruction of large scenes is considerable, often requiring hours to complete. This could prove to be inefficient in scenarios where time is of the essence.

## J  Societal Impact

Our model achieves high-fidelity 3D reconstruction. The societal impact of this development is multifaceted. On one hand, it enables significant advancements in fields such as architecture and augmented reality, improving professional practices and potentially benefiting the public by enhancing the precision and interactivity of digital models. On the other hand, the increase in computational power demands may lead to greater energy consumption, which poses environmental considerations. Moreover, there could be privacy concerns if such technology is applied to reconstruct environments from personal data without consent. Overall, while this technology presents opportunities for progress and innovation, it also requires careful consideration of ethical and environmental implications.

