# OpenReview forum: "NeuRodin: A Two-stage Framework for High-Fidelity Neural Surface Reconstruction"
_NeurIPS.cc/2024/Conference — NeurIPS 2024 poster_

### Official Review · Reviewer_EWQD · 2024-07-09

**Soundness:** 3
**Presentation:** 3
**Contribution:** 2
**Rating:** 5
**Confidence:** 3

**Summary:**

This paper focuses on improving SDF-based volume rendering and proposes a new pipeline to address issues stemming from SDF-to-density conversion and geometric regularization. First, it changes a global scale parameter to local adaptive value, allowing more flexible density values to be converted from SDF. Second, the method proposes a novel loss function to align the maximum probability distance in volume rendering with the zero-level set of the SDF representation. Third, the paper claims that SDF regularization may be too strong to allow flexible topological changes and thus proposes a two-stage training process. The coarse stage operates similarly to a density field without strong constraints, while the refinement stage then encourages enhanced smoothness.

**Strengths:**

1. The proposed improvements are well-motivated and technically sound in general.

2. The paper is well-written and easy to follow overall. It's great to have Figure 3 for illustration.

3. The paper appropriately mentions related work that shares a similar idea and the problems of prior solutions.

4. The authors compare the proposed method on two datasets and demonstrate its superiority over several baseline methods. They also provide qualitative examples for the ablation study.

**Weaknesses:**

1. It would be more convincing to include quantitative results and more qualitative examples for verifying the effectiveness of the proposed components in the ablation study. Currently, the ablation study is only conducted on a single example scene qualitatively.

2. The idea of using an adaptive scale is not quite new, as it is also seen in previous works [1]. It would be better to discuss the differences with existing works sharing a similar idea. Also, it would be better to provide more motivating examples or analysis on why an adaptive scale is important, such as when to use a large scale and when to use a small scale.

3. For the two-stage training, the motivation is to make the coarse stage more like a density field without strong constraints. The paper claims that "eliminating or downweighting any geometric constraints often results in an unnatural zero-level set." However, this point is not verified in experiments. It is also strongly recommended to add an ablation variant that does not have the eikonal loss (using estimated gradient) in the coarse stage.

4. It is common to use numerical gradients in calculating the eikonal loss. The benefit of sampling a step size should be analyzed and compared more thoroughly in the ablation studies.

5. While the variance of stochastic gradients is understandable, it is unclear why this ensures stability for large features and flexibility for complex details. More explanation and analysis would be helpful.

6. The proposed pipeline is built upon many techniques from TUVR. It would be better to directly compare with TUVR in the experiments.

7. It would be beneficial to show results on the commonly used DTU dataset as well.

8. The paper needs more careful proofreading and polishing:

 (a) Line 25: "fails to intricate geometric details"

 (b) Line 28: "is produced by is by"

 (c) Equation (11), the symbol 'n' is not explained.

 (d) Equation (8), the symbol 'd' is not explained.

[1] Wang, Zian, Tianchang Shen, Merlin Nimier-David, Nicholas Sharp, Jun Gao, Alexander Keller, Sanja Fidler, Thomas Müller, and Zan Gojcic. "Adaptive shells for efficient neural radiance field rendering." arXiv preprint arXiv:2311.10091 (2023).

**Questions:**

1. In Equation (8), why does it only penalize a single point instead of a region or multiple points (both SDF > 0 and SDF < 0)?

2. Line 273: What does "prior knowledge in terms of SDF" refer to?

3. Why would optimizing the color conditioned on the normal restrict topological change?

**Limitations:**

Yes, it's mentioned in the supplementary material.

---

> ### Author Rebuttal · Authors · 2024-08-07
>
> Thanks for your insightful feedback. For more comprehensive ablation study, comparison with TUVR, results on DTU and more explanation of stochastic gradients, please refer to our global response. Apologies for the brevity due to word constraints.
>
> ### **Motivation of Local Scale Factor**
> > Why an adaptive scale is important.
>
> We further illustrate the importance of the local scale in Figure 3 from our PDF file related to the global response, using a simple single plane scenario for explanation. This plane has a low-texture region on the left and a rich-texture region on the right.
>
> Without special constraints, the rendering weight should converge to a Dirac delta function at the surface in the richly textured region and form a scattered distribution in the weakly textured region.
>
> However, under the assumption of a global scale factor, all areas on the plane follow the same distribution which will be derived in the following sections. This means that both richly textured and weakly textured regions share the same density bias, preventing the surface from converging correctly to the richly textured surface with higher certainty.
>
> By introducing the local scale factor, the most significant difference is that the distribution of rendering weights along the ray is no longer uniform. The network can adaptively converge in the richly textured regions, and the density bias in these areas is no longer affected by the low-texture regions.
>
> In Adaptive shells, their focus is primarily on rendering quality and they did not evaluate surface reconstruction metrics. A straightforward application to surface reconstruction may lead to issues such as increased density bias. We tackle this by implementing special designs to ensure effectiveness, such as gradually scheduling the lower bound of the scale factor to correct relatively small biases and the explicit bias correction.
>
> ### **Design of Explicit Bias Correction**
> > In Equation (8), why does it only penalize a single point instead of a region or multiple points (both SDF > 0 and SDF < 0)?
>
> Although aligning $t^*$ with the SDF zero-crossing is intuitive, our experiments showed that this approach requires special design considerations. We attempted to constrain the SDF values both before and after $t^\*$, which led to very strange optimized surfaces, likely due to overly strong constraints on the SDF. We have shown the visual result in Figure 3 from our PDF file related to the global response.
>
> The key to our method's design lies in correcting large relative bias with explicit loss functions, while smaller biases are corrected by gradually scheduling the lower bound of the scale factor. We empirically found that visible surface errors are always caused by $t^\*$ is being before SDF zero-crossing points $t^0$. Therefore, we penalize the SDF value just after $t^\*$ to trend towards negative values, which prevents $t^*$ from being before $t^0$. For the case where $t^\*$ is after $t^0$, we can directly address it by gradually scheduling the lower bound of the scale factor, as the bias in this situation is relatively small.
>
> We provided a mathematical explanation under the assumption of a sufficiently small local surface. In the VolSDF framework we used, In this case, if the angle between the ray and the plane is θ, and the distance from the ray’s origin to the plane is $d$, then the distribution of the SDF along the ray can be expressed as $f(r(t)) = −t\sin θ + d$. By setting the partial derivative of the rendering weight with respect to $t$ to zero, we can directly obtain a closed-form solution for $t^\*$:
>
> $$
> t^*=\frac{s\ln (k) + d}{\sin\theta},
> $$
>
> where $k=2\sin\theta$ if $\sin \theta \leq 0.5$, and $k=(2+\sin\theta-\sqrt{\sin^2\theta+4\sin\theta})^{-1}$ if $\sin \theta > 0.5$.
>
> In this case, the closed-form solution for $t^0$ can be directly obtained as $t^0 = d/\sin\theta$. Then the distance between $t^*$ and $t^0$ is
>
> $$
>  t^{\Delta} = t^* - t^0 = \frac{\ln k }{\sin\theta}s.
> $$
>
> When $t^\Delta < 0$, $t^\*$ is before $t^0$, and when $t^\Delta > 0$, $t^\*$ is after $t^0$. In our PDF file related to the global response, we visualized the values of $t^\Delta$ under different $\theta$ in Figure 2. We found that when$t^\*$ is before $t^0$ ($\sin\theta$ less than 0.5), the relative bias is significantly greater than when $t^\*$ is after $t^0$. This aligns with our experimental findings, where visible erroneous surfaces are primarily caused by $t^*$ preceding $t^0$. Therefore, we penalize the SDF value just after $t^\*$ to prevent $t^\*$ being before $t^0$.
>
> We will include the complete derivation in the appendix.
>
> ### **Impact of Color Conditioning on Normal**
> > Why would optimizing the color conditioned on the normal restrict topological change?
>
> In Figure 6 from our PDF file related to the global response, we experimentally observed that in indoor scenes, when color conditioning on normals is applied, the optimization process becomes very slow and affects the optimization results. However, this issue is nearly absent in outdoor scenes. We believe this is primarily due to the convergence behavior of the scale factor. Indoor scenes often have many low-texture areas, leading to larger scale factors and a greater scatter distribution. Additionally, color conditioned on normals, which is intended for geometry disentanglement as mentioned in IDR and only resonable for surface points, results in many details being optimized to incorrect positions before convergence is achieved.
>
> However, our use of stochastic normals helps mitigate these erroneous surfaces. For detailed regions, the estimated normals have greater variance, which alleviates the impact of incorrect surfaces. This is also demonstrated in our ablation experiments, as shown in the table below:
>
> | F-score | D | E  | Full |
> |:-:|:-:|:-:|:-:|
> | Courthouse (outdoor) | 0.11 | 0.11 | 0.21 |
> | Meetingroom (indoor) | 0.35 | 0.21 | 0.43 |

---

> > ### Author Response · Authors · 2024-08-13
> > **Please let us know if your concerns have been addressed**
> >
> > Dear Reviewer EWQD,
> >
> > Thank you again for your review. We hope that our rebuttal could address your questions and concerns. As the discussion phase is nearing its end, we would be grateful to hear your feedback and wondered if you might still have any concerns we could address.
> >
> > Thank you for your time.

---

### Official Review · Reviewer_JYLp · 2024-07-10

**Soundness:** 3
**Presentation:** 3
**Contribution:** 3
**Rating:** 5
**Confidence:** 5

**Summary:**

The paper improved NeRF based surface reconstruction from two aspects: adaptive scale $s$ and a bias correction loss to reduce the bias, which encourages the SDF becomes negative after the maximum at $t^*$.

**Strengths:**

The position $\mathbf{r}(t)$ based scale $s$ increases the degree of freedom, which has the potential to improve the accuracy.

The bias correction loss encourages the SDF to be negative after the maximum weight point.

**Weaknesses:**

The bias correction looks like partially. It doesn't punish negative SDF before $t^*$, so it is not completed.

The experiments is not comprehensive. TUVR is not compared, which also aims to reduce the bias.

**Questions:**

While training $\mathcal{L}_{bias}$, how to evaluate $t^*$? Perhaps, it is an iterative procedure to obtain the maximum value, but can you embed the iterations into training?

How to avoid the maximum $t^*$ behind the zero iso-surface?

Why only use F-score for validation? What about other scores, such as Chamfer distance.

NeRF based training is slow, but the training times are not reported in the paper. Does it become slower due to the bias correction?

**Limitations:**

In the limitation section, it needs to more details about reconstruction of thin structures, sharp features, etc.

---

> ### Author Rebuttal · Authors · 2024-08-07
>
> Thank you for your insightful feedback. We hope the following response can address your concerns.
>
> ### **Design of Explicit Bias Correction**
> > The bias correction looks like partially. It doesn't punish negative SDF before $t^*$, so it is not completed.
>
> As you mentioned, although aligning $t^*$ with the SDF zero-crossing is intuitive, our experiments showed that this approach requires special design considerations. We tried simply constraining $f(t^\*)$ to approach zero but found that visible surface defects persisted. We also attempted to constrain the SDF values both before and after $t^\*$, which led to very strange optimized surfaces, likely due to overly strong constraints on the SDF. We have shown the visual result in Figure 3 from our PDF file related to the global response.
>
> The key to our method's design lies in correcting large relative bias with explicit loss functions, while smaller biases are corrected by gradually scheduling the lower bound of the scale factor. We empirically found that visible surface errors are always caused by $t^\*$ is being before SDF zero-crossing points $t^0$. Therefore, we penalize the SDF value just after $t^\*$ to trend towards negative values, which prevents $t^*$ from being before $t^0$. For the case where $t^\*$ is after $t^0$, we can directly address it by gradually scheduling the lower bound of the scale factor, as the bias in this situation is relatively small.
>
> We provided a mathematical explanation under the assumption of a sufficiently small local surface. In the VolSDF framework we used, In this case, if the angle between the ray and the plane is θ, and the distance from the ray’s origin to the plane is $d$, then the distribution of the SDF along the ray can be expressed as $f(r(t)) = −t\sin θ + d$. By setting the partial derivative of the rendering weight with respect to $t$ to zero, we can directly obtain a closed-form solution for $t^\*$:
>
> $$
> t^*=\frac{s\ln (k) + d}{\sin\theta},
> $$
>
> where $k=2\sin\theta$ if $\sin \theta \leq 0.5$, and $k=(2+\sin\theta-\sqrt{\sin^2\theta+4\sin\theta})^{-1}$ if $\sin \theta > 0.5$.
>
> In this case, the closed-form solution for $t^0$ can be directly obtained as $t^0 = d/\sin\theta$. Then the distance between $t^*$ and $t^0$ is
>
> $$
>  t^{\Delta} = t^* - t^0 = \frac{\ln k }{\sin\theta}s.
> $$
>
> When $t^\Delta < 0$, $t^\*$ is before $t^0$, and when $t^\Delta > 0$, $t^\*$ is after $t^0$. In our PDF file related to the global response, we visualized the values of $t^\Delta$ under different $\theta$ in Figure 2. We found that when $t^\*$ is before $t^0$ ($\sin\theta$ less than 0.5), the relative bias is significantly greater than when $t^\*$ is after $t^0$. This aligns with our experimental findings, where visible erroneous surfaces are primarily caused by $t^*$ preceding $t^0$. Therefore, we penalize the SDF value just after $t^\*$ to prevent $t^\*$ being before $t^0$.
>
> We will include the complete derivation in the appendix.
>
>
> ### **Comparison with TUVR**
> > TUVR is not compared, which also aims to reduce the bias.
>
> Since TUVR is not open-source and only reports metrics on the DTU dataset, we reproduced its unbiased SDF-to-density technique and combined it with a hash grid to create the TUVR-Grid method for comparison. Additionally, we compared our results on the DTU dataset with the reported results in the paper that did not use MVS priors (TUVR-MLP). The results are shown in the table below.
>
> Results on Tanks and Temples dataset:
> | Methods |  TUVR-Grid | Ours |
> |:---:|:---:|:--:|
> | F-score | 0.33 | 0.51 |
>
> Results on DTU dataset:
> | Methods |  TUVR-Grid | TUVR-MLP | Ours |
> |:---:|:---:|:--:|:--:|
> | CD↓ | 0.89 | 0.71 | 0.60 |
>
> The performance of TUVR is not ideal on all datasets because its unbiased nature is not fully guaranteed, and it is also somewhat affected by over-regularization.
>
> ### **Evaluation of $t^*$**
> > how to evaluate $t^*$?
>
> In lines 197-198 of the paper, we mentioned that we directly use the sampled point with the highest weight as the point where $t^\*$ is located. This is because we perform importance sampling along the ray before rendering, and the sampled points are already concentrated in areas with higher weights. Therefore, there is no need for additional iterations to find $t^*$. Experimentally, this approach does not significantly impact the results.
>
> ### **Other Evaluation Metrics**
> > Why only use F-score for validation? What about other scores, such as Chamfer distance.
>
> On the Tanks and Temples benchmark, previous work only reported the F-score as a surface quality metric. We compared our F-score results with the reported F-scores from other methods, although we also evaluated additional metrics using our method. However, these additional metrics cannot be directly compared with other methods.
>
> Additionally, we compared the PSNR metric for image reconstruction on the Tanks and Temples dataset and the Chamfer distance metric on the DTU dataset. The results are shown in the table below:
>
> | Methods | NeuS |  Neuralangelo-22 | Ours-19 | Ours-22 |
> |:-------:|:----:|:----:|:-------:|:-------:|
> | PSNR    |   24.58   |   27.24   | 26.90   | 27.67   |
>
> | Methods | NeuS |  TUVR-Grid | TUVR-MLP | Neuralangelo | Ours |
> |:---:|:---:|:---:|:--:|:--:|:--:|
> | CD↓ | 0.84 | 0.82 | 0.71 | 0.61 | 0.60 |
>
> ### **Impact of Explicit Bias Correction on Training Time**
> > Does it become slower due to the bias correction?
>
> After incorporating the explicit unbias correction design, the time per iteration increased from approximately 59ms to 63ms. After 300,000 iteration, this results in an additional 20 minutes of training time. This is primarily because we need additional network inference to compute the SDF slightly beyond $t^*$.

---

> > ### Comment · Reviewer_JYLp · 2024-08-11
> >
> > I appreciate the authors' feedback. After reading the rebuttal and other reviews, I am glad to increase my score.

---

> > > ### Author Response · Authors · 2024-08-11
> > >
> > > Thank you for your thoughtful consideration. I truly appreciate your willingness to increase your score!

---

### Official Review · Reviewer_9bjU · 2024-07-10

**Soundness:** 3
**Presentation:** 3
**Contribution:** 3
**Rating:** 6
**Confidence:** 4

**Summary:**

This paper introduces an innovative two-stage framework, NeuRodin, for neural surface reconstruction that significantly improves upon previous SDF-based methods. Locally adaptive parameters for SDF-to-density conversion and a novel explicit bias correction are introduced to enhance the fine reconstruction of SDF surfaces.

**Strengths:**

1. The paper thoroughly elaborates on the shortcomings of current SDF-based approaches and provides insightful and impactful plug-and-play solutions, such as SDF-to-density conversion and explicit bias correction.
2. The paper is clearly written with detailed explanations of the methodology, the challenges addressed, and the solutions proposed.
3. Strong experimental results demonstrate that the proposed framework can generate high-quality surface reconstruction results.

**Weaknesses:**

1. The two-stage optimization process proposed in the paper seems somewhat cumbersome, and finding a balance between over-regularization and fine reconstruction is challenging. This detailed design somewhat weakens the depth of discussion on the essential issues of SDF reconstruction. Easily portable modification modules, such as explicit bias correction and SDF-to-density conversion, could have a more far-reaching impact.
2. The difference between the ideal situation discussed by the authors in Explicit Bias Correction and the SDF distribution in actual training seems to have a similar mechanism to the convergence difficulty of shape adjustment discussed in [1]. Is it possible to avoid the biased situation shown in Figure 3 by introducing depth or other supervision information?
3. In the experimental section, Table 2 does not explain the measurement indicator of the data. Combined with Appendix 5, it can be inferred that the indicator is the percentage F-Score.
4. There is a lack of comparison of image reconstruction results. Since color is used in the training process and the baseline method in the paper includes image reconstruction comparisons, why does this paper not provide qualitative or quantitative results of image reconstruction?

[1] Yang, Huizong, et al. "Stabilizing the Optimization of Neural Signed Distance Functions and Finer Shape Representation." *Advances in Neural Information Processing Systems* 37 (2023).

**Questions:**

Refer to the weaknesses section.

**Limitations:**

The authors discuss the limitations of the proposed model.

---

> ### Author Rebuttal · Authors · 2024-08-07
>
> Thank you for your insightful feedback. The issues we mentioned regarding SDF-based rendering, such as the bias problem, might indeed be addressed through improved SDF-to-density modeling. While it is possible to tackle these issues directly through mathematical modeling, we believe that our detailed design can be very easily implemented within the existing framework and effectively and significantly improve surface quality. This also provides considerable value.
>
> ### **Impact of Additional Signal Supervision**
> > Is it possible to avoid the biased situation shown in Figure 3 by introducing depth or other supervision information?
>
> We tested monocular depth and normal supervision on an indoor scene from ScanNet++ to examine the density bias. We found that the density bias no longer caused significant surface errors. This aligns with intuition, as the impact of density bias depends on the convergence of the surface. Any SDF-to-density design ensures that when the scale factor approaches zero, the weights along the ray can approximate a Dirac distribution at the SDF zero-crossing points.
>
> We believe that the bias issue is caused by photometric ambiguity, meaning that the surface cannot be accurately determined based on color alone. This is consistent with the outlier distribution in MVS methods. From the MVS perspective, the scattered distribution in low-texture areas of MVS results is similar to the weight distribution along the ray and is considered outliers and removed. However, our goal is dense surface reconstruction, and we need to uniquely determine the surface within this scattered distribution. Therefore, we select the part with the highest weight as the most reliable surface.
>
> When we have more reliable supervision signals, such as monocular depth and normals, these outliers decrease because the model can use these additional signals to accurately determine the surface.
>
> ### **Image Reconstruction Comparision**
> > There is a lack of comparison of image reconstruction results. Since color is used in the training process and the baseline method in the paper includes image reconstruction comparisons, why does this paper not provide qualitative or quantitative results of image reconstruction?
>
> While our design is primarily intended for surface reconstruction tasks, we have also compared the results of NeuS and Neuralangelo using the image quality evaluation methods from Neuralangelo, across different parameter scales.
>
> | Methods | NeuS |  Neuralangelo-22 | Ours-19 | Ours-22 |
> |:-------:|:----:|:----:|:-------:|:-------:|
> | PSNR    |   24.58   |   27.24   | 26.90   | 27.67   |
>
> As shown, with fewer parameters, our method (Ours-19) achieves image reconstruction quality close to that of Neuralangelo. With a larger number of parameters, our method (Ours-22) surpasses Neuralangelo in terms of image reconstruction quality.
>
> ### **Addressing Typographical Errors and Omissions**
> > In the experimental section, Table 2 does not explain the measurement indicator of the data. Combined with Appendix 5, it can be inferred that the indicator is the percentage F-Score.
>
> Thank you for your valuable feedback and for pointing out the typographical errors and missing elements in our manuscript. We sincerely apologize for these mistakes. We will carefully revise the manuscript to correct these issues and ensure that it meets the highest standards of clarity and accuracy.

---

> > ### Comment · Reviewer_9bjU · 2024-08-10
> >
> > After reviewing the authors' responses and supplementary materials, I appreciated the additional experiments and explanations provided during the rebuttal phase. The concerns regarding the supervision of the additional signal were addressed, and the results of the rendering of the images demonstrated the effectiveness of the method, prompting me to improve my initial score.

---

> > > ### Author Response · Authors · 2024-08-11
> > >
> > > Thank you very much for your prompt reply and for increasing your score!

---

### Official Review · Reviewer_o1gj · 2024-07-14

**Soundness:** 3
**Presentation:** 3
**Contribution:** 3
**Rating:** 6
**Confidence:** 2

**Summary:**

This article introduces NeuRodin, a Signed Distance Function (SDF)-to-density-based neural surface reconstruction method. The author summarizes the two main factors, SDF-to-density representation and geometric regularization, which cause low-quality performance in SDF-based methods and improve them in the pipeline. They argue that the widely used global scale parameter may cause the identical density in the same SDF level set. The paper attempts to solve this problem by using a non-linear mapping to obtain this scale parameter and adding a bias of density.  For the geometric regularization, they mentioned some reasons such as the Eikonal loss and smoothness constraints that caused oversmoothness. Stochastic-step numerical gradient estimation and two-step training strategy are employed for over-regularization issue. The results overperform some baselines and ablation studies show the effectiveness of the modules.

**Strengths:**

The mentioned factors theoretically may cause the low-quality performance. The pipeline is clear, and theories are reasonable.
The strategies and improvements enhance the performance of the model.
The shown depth maps examples also improved with the proposed strategies. This could be useful for other depth estimation tasks.

**Weaknesses:**

The authors only provide F-score for the evaluation metrics except for some single scenes in appendix. It will be good to provide more such as PSNR, SSIM, etc.
There are some minor mistakes in Table 5. The underline format in caption is not consistent with the italics format in the table.
Is it possible to compute the proposed method under gird resolution of 2^22 just as nerualangelo did? This could be a fairer comparison in table 1.
It will be good to discuss the runtime compared with other baselines.

**Questions:**

Is it possible to compute the proposed method under gird resolution of 2^22 just as nerualangelo did? This could be a fairer comparison in table 1.
It will be good to discuss the runtime compared with other baselines.

**Limitations:**

The authors only provide F-score for the evaluation metrics except for some single scenes in appendix. It will be good to provide more such as PSNR, SSIM, etc.

---

> ### Author Rebuttal · Authors · 2024-08-07
>
> Thank you for your insightful feedback. Below, we address the concerns raised in the review.
>
>
> ### **Image Reconstruction Comparision**
> > The authors only provide F-score for the evaluation metrics except for some single scenes in appendix. It will be good to provide more such as PSNR, SSIM, etc.
>
> While our design is primarily intended for surface reconstruction tasks, we have also compared the results of NeuS and Neuralangelo using the image quality evaluation methods from Neuralangelo, across different parameter scales.
>
> | Methods | NeuS |  Neuralangelo-22 | Ours-19 | Ours-22 |
> |:-------:|:----:|:----:|:-------:|:-------:|
> | PSNR    |   24.58   |   27.24   | 26.90   | 27.67   |
>
> As shown, with fewer parameters, our method (Ours-19) achieves image reconstruction quality close to that of Neuralangelo. With a larger number of parameters, our method (Ours-22) surpasses Neuralangelo in terms of image reconstruction quality.
>
> ### **Hash Grid with a Dictionary size of $2^{22}$**
> >  Is it possible to compute the proposed method under gird resolution of 2^22 just as nerualangelo did?
>
> We also tested our method on the Tanks and Temples dataset with a hash dictionary size of 2^22 as shown in the table below.
>
> | Methods | Neuralangelo-19 | Neuralangelo-22 | Ours-19 | Ours-22 |
> |:-------:|:---------------:|:---------------:|:-------:|:-------:|
> | F-score    |   0.43   |   0.50   |         0.51        |  0.50  |
>
> We found that increasing the number of parameters did not significantly improve the results but instead dramatically increased the running time (from 18 hours to 48 hours). The redundant parameters (with a dictionary size of $2^22$ resulting in over a hundred million parameters) did not enhance surface accuracy but did help fit the training images better (PSNR from 26.90 to 27.67). Neuralangelo addresses some surface defects through over-parameterization. However, we achieve the same effect with significantly fewer parameters, thanks to our specialized design.
>
> Furthermore, we test with a hash dictionary size of $2^{22}$ in a large-scale scenario from BlendedMVS dataset. By significantly increasing the number of parameters, the model is able to capture fine-grain details on images, thereby preserving more details in the reconstructed surface. In our PDF file related to the global response, Figure 1 displays the quantitative comparison.
>
> ### **Training time**
> > It will be good to discuss the runtime compared with other baselines.
>
> As mentioned in lines 504-506 of the original text, on the Tanks and Temples dataset, Neuralangelo requires nearly 48 GPU hours for optimization, whereas our method only requires approximately 18 GPU hours. This is due to our specialized design for addressing surface defects, which allows us to maintain performance without requiring as many parameters as Neuralangelo.

---

> > ### Author Response · Authors · 2024-08-13
> > **Please let us know if your concerns have been addressed**
> >
> > Dear Reviewer o1gj,
> >
> > Thank you again for your review. We hope that our rebuttal could address your questions and concerns. As the discussion phase is nearing its end, we would be grateful to hear your feedback and wondered if you might still have any concerns we could address.
> >
> > Thank you for your time.

---

### Author Rebuttal · Authors · 2024-08-07

We express our sincere gratitude to all the reviewers for their valuable insights and constructive feedback on our work. We truly appreciate your dedicated efforts and the time you have devoted to evaluating our work.
Here we address some common concerns raised by reviewers. We also have provided a PDF for visualizing results or charts to assist in illustrating and explaining the issues. Please check.
### **Image Reconstruction Comparision**
While our design is primarily intended for surface reconstruction tasks, we have also compared the results of NeuS and Neuralangelo using the image quality evaluation methods from Neuralangelo, across different parameter scales.

| Methods | NeuS |  Neuralangelo-22 | Ours-19 | Ours-22 |
|:-------:|:----:|:----:|:-------:|:-------:|
| PSNR    |   24.58   |   27.24   | 26.90   | 27.67   |

As shown, with fewer parameters, our method (Ours-19) achieves image reconstruction quality close to that of Neuralangelo. With a larger number of parameters, our method (Ours-22) surpasses Neuralangelo in terms of image reconstruction quality.


### **Comparison with TUVR**
Since TUVR is not open-source and only reports metrics on the DTU dataset, we reproduced its unbiased SDF-to-density technique and combined it with a hash grid to create the TUVR-Grid method for comparison. Additionally, we compared our results on the DTU dataset with the reported results in the paper that did not use MVS priors (TUVR-MLP). The results are shown in the table below.

Results on Tanks and Temples dataset:
| Methods |  TUVR-Grid | Ours |
|:-:|:-:|:-:|
| F-score | 0.33 | 0.51 |

Results on DTU dataset:
| Methods |  TUVR-Grid | TUVR-MLP | Ours |
|:-:|:-:|:-:|:-:|
| CD↓ | 0.89 | 0.71 | 0.60 |

The performance of TUVR is not ideal on all datasets because its unbiased nature is not fully guaranteed, and it is also somewhat affected by over-regularization.

### **Comparison on DTU Benchmark**
Although our design is not specifically tailored for single-object datasets, we validated our method on the DTU Benchmark. The results are shown in the table below. We found that, even without special parameter tuning, our method achieves results comparable to Neuralangelo and surpasses other baseline methods such as TUVR.

| Methods | NeuS |  TUVR-Grid | TUVR-MLP | Neuralangelo | Ours |
|:-:|:-:|:-:|:-:|:-:|:-:|
| CD↓ | 0.84 | 0.82 | 0.71 | 0.61 | 0.60 |

### **More Ablation Study**

We conducted additional ablation experiments on the Tanks and Temples training set and we included two more cases to verify the role of Eikonal loss in maintaining the natural zero level set during the first stage, as well as the impact of color condition on normals.

The five cases are:

A: Without the local scale factor

B: Changing the stage 1 estimated gradient to the analytical gradient

C: Without explicit bias correction

D: Without stage 1 Eikonal loss, but with color conditioning on the estimated normal

E: Without stage 1 Eikonal loss, but with color conditioning on the analytical normal

The quantitive results are as follows:

| Case    | A    | B    | C |  D   | E    | Full Model |
|:-:|:-:|:-:|:-:|:-:|:-:|:-:|
| F-score | 0.49 | 0.42 | 0.49 | 0.42 | 0.40 | 0.51       |

The results from cases A, B, and C demonstrate the effectiveness of the techniques we proposed. The results from cases D and E indicate the necessity of applying Eikonal loss during the first stage.

Here, we present the results of removing the Eikonal loss during the first stage (cases D and E), which numerically demonstrate the necessity of this loss in the first stage. If removed, the following issues arise:

1. Noisy surfaces behind the correct surface.
2. Some large-scale areas display challenging-to-fill holes in the subsequent stage.
3. Under this poor SDF initialization in stage one, the optimization in stage 2 is not rubust.

### **Training time**
As mentioned in lines 504-506 of the original text, on the Tanks and Temples dataset, Neuralangelo requires nearly 48 GPU hours for optimization, whereas our method only requires approximately 18 GPU hours. This is due to our specialized design for addressing surface defects, which allows us to maintain performance without requiring as many parameters as Neuralangelo.

### **More Explanation of Stochastic Gradients**

As discussed in the main paper, our stochastic gradient estimation introduces some uncertainty into the true normals. For large-scale features, the Eikonal loss with varying step sizes can still be successfully minimized (since SDF near a large-scale plane should satisfy the Eikonal equation for different step sizes). In other words, the variance of stochastic gradients is small for large-scale features, making it easier for the model to minimize the Eikonal loss. However, for fine details, the random step sizes lead to high variance in the estimated normals, which reduces the impact of the Eikonal loss and makes it more challenging for the model to minimize samples with high variance.

---

### Decision · Program_Chairs · 2024-09-25

**Decision:**

Accept (poster)

**Comment:**

This paper makes a clear incremental contribution to SDF volume rendering. The discussion with the authors and reviewers efficiently clarified the points raised in the reviews and an acceptance consensus was reached.